# Lack of SMARCB1 expression characterizes a subset of human and murine peripheral T-cell lymphomas

Peripheral T-cell lymphoma, not otherwise specified (PTCL-NOS) is a heterogeneous group of malignancies with poor outcome. Here, we identify a subgroup, PTCL-NOS[SMARCB1-], which is characterized by the lack of the SMARCB1 protein and occurs more frequently in young patients. Human and murine PTCL-NOS[SMARCB1-] show similar DNA methylation profiles, with hypermethylation of T-cell-related genes and hypomethylation of genes involved in myeloid development. Single-cell analyses of human and murine tumors revealed a rich and complex network of interactions between tumor cells and an immunosuppressive and exhausted tumor microenvironment (TME). In a drug screen, we identified histone deacetylase inhibitors (HDACi) as a class of drugs effective against PTCL-NOS[Smarcb1-]. In vivo treatment of mouse tumors with SAHA, a pan-HDACi, triggered remodeling of the TME, promoting replenishment of lymphoid compartments and reversal of the exhaustion phenotype. These results provide a rationale for further exploration of HDACi combination therapies targeting PTCL-NOS[SMARCB1-] within the TME.

PTCL-NOS is among the most common forms of mature T-cell lymphoma[1,2]. It comprises a heterogeneous group of aggressive malignancies that predominantly affect adults[3] and less frequently children, adolescents and young adults (CAYA) below the age of 25 years[4]. Because patients respond poorly to current treatment regimens, identification of new therapeutic strategies is required. According to the 5th edition of the WHO classification of lymphoid neoplasms, PTCL-NOS represents a heterogeneous diagnostic category that is differentiated from, e.g., nodal T-follicular helper cell lymphoma[5]. High expression of either *TBX21* or *GATA3* characterizes two molecular variants PTCL-TBX21 and PTCL-GATA3, indicating programs in T1 and T2 helper cells, respectively[6]. Epigenetic mechanisms play a particularly important role in the pathogenesis of PTCL-TBX21, since DNA or histone methylation genes are often mutated in this entity[7,8].

Chromatin remodeling genes, such as *SMARCA4*, *ARID1A*, and other members of the SWI/SNF complex, can also be mutated in different lymphomas[9–14]. SMARCA4 and ARID1A form part of the human SWI/SNF complex BAF (BRG1/BRM-associated factor), which mobilizes nucleosomes along the DNA[15,16]. The loss of SMARCB1, another BAF

subunit, generally results in reduced chromatin accessibility and transcriptional repression[17]. The biallelic inactivation of this tumor-suppressor gene is intimately linked to the development of pediatric embryonal cancer and rhabdoid tumors[18], while mono-allelic inactivating mutations in the germline are the molecular basis of the rhabdoid tumor predisposition syndrome type 1[19,20]. SMARCB1 has key roles in human lymphocyte development and function[21], but little is known about its role in lymphoma pathogenesis, despite *SMARCB1* deletions and mutations described in T-cell prolymphocytic leukemia and cutaneous T-cell lymphoma[22–24] and, recently, biallelic *SMARCB1* loss was also associated with aggressive hematopoietic malignancy[25]. In a genetic mouse model (CD4-Cre *Smarcb1*[fl/fl]) Smarcb1 inactivation in mature T-cells triggers the development of oligoclonal TdT−, TCR+, CD3+, CD8+, and CD4− mature PTCL[26] and, rarely, also rhabdoid tumors[27].

Here we describe a subgroup of human PTCL, referred to as PTCL-NOS[SMARCB1−], characterized by the loss of SMARCB1 and predominantly affecting younger individuals. Through comparative epigenomic studies of human PTCL-NOS[SMARCB1−] and murine tumors from the Smarcb1-knockout model, we characterize common pathways of

✉e-mail: Kornelius.kerl@ukmuenster.de

lymphomagenesis in these tumors. We further describe the transcriptional landscape of this entity at the single-cell level and investigate the functional interaction network between lymphoma cells and their microenvironment. Finally, we show that SAHA (suberoylanilide hydroxamic acid/vorinostat), a pan-HDACi, is able to remodel the cellular immune landscape of PTCL-NOS[Smarcb1−] in a favorable manner and thus could be a useful therapeutic agent in this type of cancer.

## Results

### A SMARCB1-negative subgroup of PTCL-NOS is predominant in younger patients

In a mouse tumor model, a reversible conditional *Smarcb1* allele causes the majority of mice to develop a mature T-cell lymphoma within a few weeks upon loss of Smarcb1[27]. Building on this observation, we evaluated SMARCB1 expression in different human mature T-cell lymphoma cohorts (Fig. 1A; Suppl. Fig. 1).

We first investigated T-cell prolymphocytic leukemia (T-PLL), a mature T-cell leukemia. SMARCB1 gene loss was present in 2 out of 16 T-PLL cases as well as in the T-PLL-like cell line SUP-T11 (Table S1). Gene expression and DNA methylation data revealed no significant difference in SMARCB1 expression and promoter methylation in T-PLL samples compared to non-malignant T-cells[28] (Suppl. Fig. 2). SMARCB1 protein expression was confirmed in lysates of T-PLL patients and in SUP-T11 cells (Suppl. Fig. 2).

We next performed immunohistochemical analysis of SMARCB1 expression in 15 cases of mycosis fungoides (MF), a type of cutaneous T-cell lymphoma. Only one case displayed a few scattered negative elements with irregular nuclear profiles located in the superficial dermis (5%). All the remaining MF cases showed indistinctively intense nuclear staining (Table S2; Suppl. Fig. 3). Immunohistochemical analysis of SMARCB1 expression in intestinal lymphomas (enteropathy-associated T-cell lymphoma, EATL, and monomorphic epitheliotropic intestinal T cell lymphoma, MEITL) also showed SMARCB1 positivity in the majority of samples (Table S3; Suppl. Fig. 3). As we did not detect common loss of SMARCB1 expression in T-PLL, MF, EATL and MEITL, these entities do not appear to be the human counterpart to the mature T-cell lymphomas in the Smarcb1-deficient mouse model.

Finally, we examined transcriptomic data of 225 mature T-cell lymphomas from the TENOMIC database[29–35], including 76 cases of PTCL-NOS and 100 cases of angioimmunoblastic T-cell lymphoma (AITL) as well as 19 Natural killer cell (NK)/T-cell lymphomas, 11 hepatosplenic T-cell lymphomas and 19 ALK-negative anaplastic T-cell lymphomas (ALK-ALCL). SMARCB1 gene expression was heterogeneous within the different entities (Fig. 1B). Nevertheless, very low SMARCB1 expression levels were significantly more frequent in the PTCL-NOS than in the AITL group (Fig. 1B; adjusted $p < 0.0001$, Wilcoxon test). Therefore, we focused on PTCL-NOS in further assessment. SMARCB1 protein expression was examined in selected PTCL-NOS in the TENOMIC dataset, as well as in additional adults (over 25 years) and CAYA (Fig. 1C, G; Table S4). While in the original screening cohort, only 1 out of 28 (3.6%) adult PTCL-NOS patients was negative for SMARCB1 staining, this number increased to 4 out of 13 cases (31%) in the CAYA age group (Fig. 1D; $p < 0.05$, Fisher's exact test). This was an unexpected finding given the fact that adult samples selected for protein analysis show the lowest SMARCB1 RNA expression. The extended cohort included cases specifically selected for SMARCB1 protein loss ($n = 5$). Combining both cohorts revealed an even increased enrichment of SMARCB1-negative cases in CAYA patients (47% compared to 7% in adult patients, $p = 0.0026$, Fisher's exact test). In both cohorts SMARCB1 protein deficiency significantly correlated with younger age in PTCL-NOS (Fig. 1E).

### Molecular characterization of SMARCB1-negative PTCL-NOS

We performed RNA profiling of three SMARCB1-positive and three SMARCB1-negative PTCL-NOSs. Comparing gene expression of common T cell genes did not reveal a specific pattern of the PTCL-GATA3 or PTCL-TBX21 subtype. (Fig. 1F). Whereas one SMARCB1-negative case showed characteristics of cytotoxic PTCL-NOS (CD8+, patient 5), the other was CD4/8-positive (patient 1) arguing for a unique subtype of SMARCb1-deficient PTCLs. All cases were Tdt and CD30 negative and showed expression of at least one T cell marker (Fig. 1G).

Next, we addressed potential molecular explanations for the lack of SMARCB1 expression in human PTCL-NOS. Genomic profiling of nine SMARCB1-negative CAYA samples via (targeted) NGS ($n = 9$) and OncoScan array ($n = 7$) showed biallelic mutations/deletion in three cases (Fig. 1H–J). In one case biallelic loss was confirmed using FISH (Table S5; Suppl. Fig. 4). One additional case showed an exon 1 loss leading to SMARCB1 absence on protein level (Suppl. Fig. 4). Additionally, three cases showed heterozygous mutations or copy number alterations in *SMARCB1*, while two cases did not show any genetic alterations (Fig. 1J; Table S5).

As genomic profiling did not provide conclusive evidence for biallelic mutation as common cause of SMARCB1 inactivation, which is typical for rhabdoid tumors, we next investigated DNA methylation as an alternative mechanism for gene silencing. The DNA methylation profile revealed higher SMARCB1 promoter methylation in four SMARCB1-negative PTCLs, including two cases without genomic alterations, compared to normal T cells and other malignant T cell populations (Wilcoxon test: $p < 0.001$ and $p < 0.01$, respectively) (Table S6; Suppl. Fig. 5). We conclude that loss of SMARCB1 expression in human PTCL largely occurs via somatic mutation and/or epigenetic silencing, whereas germline SMARCB1 mutations have not yet been observed in all samples tested. Moreover, no specific loss of other SWI/SNF member genes was detected in PTCL-NOS (Suppl. Fig. 5).

For further insight into the molecular properties of this PTCL subtype, we established a mouse model by inactivating murine Smarcb1 in mature T cells using Cd4-cre::*Smarcb1*[fl/fl] mice[26]. These mice develop an enlarged spleen and a concomitant loss of red/white pulp organization with 100% penetrance after 9 to 12 weeks (Table S7; Suppl. Fig. 6). We then recorded the DNA methylation profiles of these murine tumors ($n = 5$) as well as those of human SMARCB1-negative PTCL-NOS samples ($n = 5$) with non-neoplastic CD3 + T cells (Table S8). The tumor DNA was globally hypomethylated compared to that from non-malignant T cells (Fig. 2A, B). Comparison between human and murine tumors showed similar proportions of differentially hypo- and hypermethylated gene loci (hypo: mouse 34.9%, human 23.4%; hyper: mouse 65.1%, human 76.6%, t-test: σ/σmax > 0.4, q < 0.01).

Given the similar proportions of differentially hypo- and hypermethylated loci in both species, the question arose if the same biological processes and pathways are affected. Within these differently methylated loci we found 104 genes concordantly hypomethylated and 534 genes concordantly hypermethylated in human and murine lymphomas (Fig. 2C; Supplementary Data S1). Remarkably, gene ontology analyses revealed an enrichment of genes concordantly linked to hypomethylated CpGs involved in myeloid leukocyte differentiation (e.g., *CDK6*) (Fig. 2D; Supplementary Data S1). By using a more stringent false discovery rate (t-test: σ/σmax > 0.4, q < 1e-5), the number of concordantly hypermethylated genes in both species was reduced to 28 genes covering a network around cancer/T-cells related genes (e.g., *CTLA4, ETS1*) (Supplementary Data S2). Genes linked to concordantly hypermethylated CpGs are significantly enriched in regulators of myeloid apoptosis and lymphocyte differentiation (Fig. 2D; Table S9). Taken together, this suggests that SMARCB1 dysfunction in these tumors is associated with lineage infidelity and/or plasticity of the lymphoid compartment.

### The cellular and transcriptomic landscape of SMARCB1-negative PTCL-NOS

To shed more light on this putative PTCL subtype, we used single-cell RNA sequencing (scRNA-seq). Tumor material was isolated from

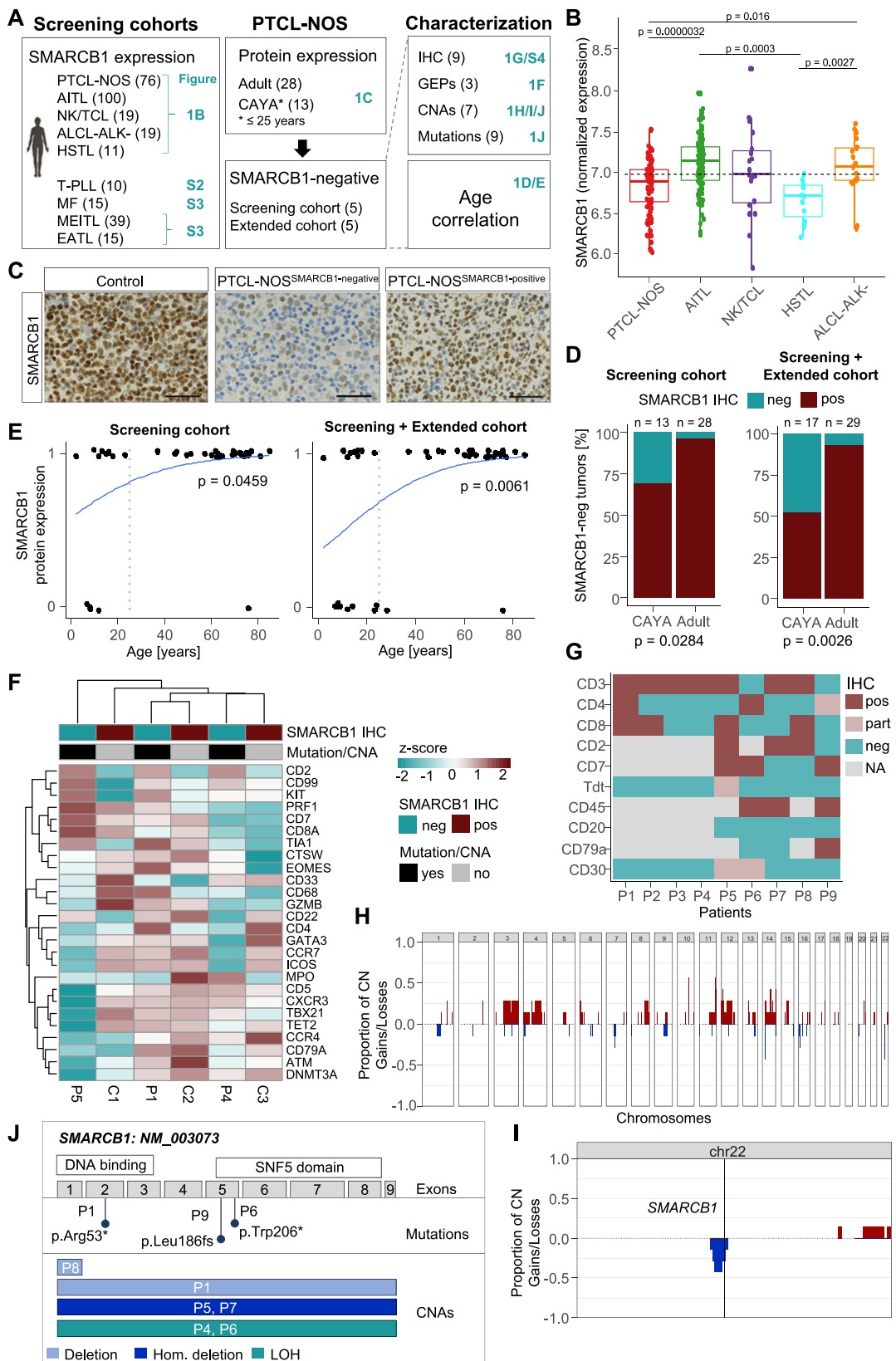

archival formalin-fixed, paraffin-embedded (FFPE) blocks from three female and two male patients aged between 7 and 12 years and subjected to scRNA-seq using Chromium Fixed RNA Profiling technology from 10X Genomics (Fig. 3A, Table S10). After integrating the five individual data sets, a total of 19,678 single cells were assigned to 19 different clusters (Fig. 3B; Suppl. Fig. 7A). Differentially expressed

genes (DEG) analysis was then performed and cluster-specific cell types annotated through an interplay of bioinformatic analyses and manual curation (Suppl. Methods; Supplementary Data S3). Tumor cells were differentiated from non-malignant cells of the tumor microenvironment (TME) using the following five criteria: (i) negligible SMARCB1 expression, (ii) CD2 positivity (as a marker of the

**Fig. 1 | Thirty-one percent of PTCL-NOS are SMARCB1-negative in pediatric and young patients. A** Overview of patient cohorts and methods for genetic characterization of patients with T-cell lymphomas and exact patient number for each cohort. **B** SMARCB1 RNA expression in samples from the TENOMIC study (*n* = 225). Normalized expression is shown. The dashed line represents the median expression value of all subgroups. Wilcoxon test (two-sided), all significant adjusted *p* values (Benjamini–Hochberg) are indicated. Boxplot settings: middle, median; lower hinge, 25% quantile; upper hinge, 75% quantile; upper/lower whisker, largest/smallest observation less/greater than or equal to upper/lower hinge ±1.5 × IQR. **C** Immunohistochemistry of SMARCB1. Exemplary images of sections from SMARCB1-positive and negative human PTCL-NOS cases compared to control tissue (tonsils). Scale bars: 50 μm. The experiment was performed for five SMARCB1-negative lymphomas. **D** 31% of PTCL-NOS patients under 25 years old (CAYA) (*n* = 4/13) and 3.6% of adults (*n* = 1/28) present loss of SMARCB1 protein expression. Fisher's exact test (two-sided), *\*p* = 0.0284. Adding the extended cohort, 47% (*n* = 8/17) of CAYA patients and 7% (*n* = 2/29) of adults were negative for SMARCB1 expression. Fisher's exact test, *\*p* = 0.0026. Protein expression was evaluated using IHC. **E** Correlation of SMARCB1 protein expression and age in PTCL-NOS patients. Protein expression was evaluated using IHC. Negative cases with no SMARCB1 expression are labeled with '0' while cases with complete or partial SMARCB1 expression are labeled with 1. Data is shown for 42 patients (14 CAYA patients, 28

adults). Wald test from binomial generalized linear model (two-sided), *\*p* value = 0.0459. Adding the extended cohort, the *p* value decreases to 0.0061. **F** Transcriptomic profiling of three SMARCB-negative PTCL-NOS patient samples (patient 1, 4, and 5) and three control SMARCB1-positive PTCL-NOS samples (C1-3). HTG transcriptome panel was used and normalized gene expression is shown for genes connected to PTCL-NOS subtypes. **G** Immunohistochemical characterization of nine SMARCB1-negative cases. **H, I** Copy number profiling of seven SMARCB1-negative cases using OncoScan. The proportion of gains and losses is shown for all autosomes (**H**) and chromosome 22 in detail (**I**). **J** Summary of copy number and mutational profiling in nine SMARCB1-negative cases. Source data of **B**, **D** and **E** are provided as a Source Data file. **A** Created with BioRender.com released under a Creative Commons Attribution-NonCommercial-NoDerivs 4.0 International license. PTCL-NOS Peripheral T cell lymphoma not otherwise specified, AITL Angioimmunoblastic T cell lymphoma, NKTCL Natural killer/ T cell lymphoma, HSTL Hepatosplenic T cell lymphoma, ALCL-ALK- ALK-negative anaplastic large cell lymphoma, T-PLL T-cell prolymphocytic leukemia, MF mycosis fungoides, MEITL monomorphic epitheliotropic intestinal T cell lymphoma, EATL enteropathy-associated T-cell lymphoma, CAYA children adolescents and young adults, IHC immunohistochemistry, GEPs gene expression profiles, CNAs copy number alterations, pos positive, neg negative, part partial expression, P1-9 patient 1–9, hom homozygous, LOH loss of heterozygosity.

---

mature T-cell origin of PTCL); (iii) EZH2 positivity (frequently over-expressed in PTCL-NOS[36], (iv) KIT positivity (based on our previous observations) and (v) high proliferative activity as exemplified by strong expression of MKI67 (Suppl. Fig. 7B). Five clusters met these criteria. The remaining clusters were classified as T-cells, B- and plasma B-cells, monocytes/macrophages (Mono/Mac), OSCAR+ osteoclasts (OCL), plasmacytoid and conventional type 1 dendritic cells (pDC and cDC1) and LAMP3+ mature DCs enriched in immunoregulatory molecules (mregDC). In addition, we identified a rich compartment of non-hematopoietic cells (NHC), which matched with a single-cell atlas of stromal cells in human lymph nodes and lymphoma[37]. These included blood and lymphatic endothelial cells (BEC and LEC), pericytes (PC), non-endothelial stromal cells (NESC) as well as CCL19+/CCL21+ fibroblastic reticular cells (FRC) (Fig. 3C; Suppl. Fig. 7C). The heterogeneity of the Tumor/T-cell as well as the myeloid Mono/Mac compartment (hereafter Myeloid) were examined at higher resolution by separation and re-clustering of the corresponding subsets. This procedure resulted in 13 new clusters T0-T12 for the first subset, eight of which were identified as EZH2-positive tumor cell clusters (gray color code) and the rest as T-cells (green color code) (Fig. 3D). From the two initial Myeloid clusters, six new clusters M0–M5 emerged (Fig. 3E).

Functional gene expression programs were determined via alignment with recently described cancer hallmark metaprograms (MPs)[38]. A number of tumor cell clusters showed clear matches to various MPs, whereas the T-cell clusters behaved inconspicuously (Fig. 3F, Supplementary Data S4, 5). T5 was associated with cell cycle programs (e.g., 49 hits of T5 DEGs on the 50 signature genes of MP1_G2/M), so we refer to it as "Cycling" hereafter (Fig. 3G). T1 showed 33/50 hits on MP20_MYC (i.e., oncogenic MYC signaling; hereafter "T1_MYC"), and cluster T9 showed 36/50 hits on MP12_EMT I (i.e., epithelial-to-mesenchymal transition; hereafter "T9_EMT"). Within the Myeloid subset, a dichotomous distribution was observed: cluster M3 was assigned as Cycling and cluster M0 as EMT, while clusters M2, M4, and M5 each showed the strongest agreement with the metaprogram MP5_Stress (Fig. 3H, Supplementary Data S6, 7). Apparently, functional metaprograms manifest in surprisingly similar ways in developmentally unrelated cell populations of tumor and TME, suggesting extensive communication within the local tumor niche. This is further illustrated by the violin plot in Suppl. Fig. 8A, which shows that subpopulations of tumor and myeloid cells express identical signature genes of the Cycling, MYC, and EMT programs. Gene ontology (GO) annotation and gene set enrichment analysis (GSEA) of the T9/M0 EMT

program confirmed the central role of the consensus signature genes in the organization of the extracellular matrix (ECM) in the tumor niche (Suppl. Fig. 8B–D). A functional gene network analysis was performed for the shared stress program of M2, M4 and M5, which revealed that it is associated with tumor necrosis factor (TNF) signaling via NFKB and clusters around a central core of the AP-1 (JUN/FOS) transcription factor family (Suppl. Fig. 8E, F).

Next, we focused on a thorough analysis of the phenotypic characteristics and functional states of individual clusters. Immune gene profiling of the Tumor/T-cell compartment revealed the absence of innate lymphoid cells. We found neither significant expression of cell type-defining marker genes for NK cells (CD56 and/or CD16) nor for mucosal-associated invariant T (MAIT) cells. The five T-cell clusters could be resolved into the CD4/CD8 double-negative (DN) clusters T0 and T7, the CD8-positive clusters T3 and T10, and a single CD4-positive cluster T11 (Fig. 3I). Strikingly, three of the five clusters expressed neither CD3 genes nor LCK, a T-cell receptor (TCR) complex-associated kinase with an important function in TCR signaling. We, therefore, define these clusters as phenotypically compromised. The functional state of the individual Tumor/T-cell clusters was further evaluated by comparison with a single-cell reference atlas of tumor-associated T-cells[39]. The majority (6/8) of the tumor cell clusters showed a precursor-exhausted (Tpex)-like expression profile characterized as PD-1 (PDCD1)/TIGIT[high] and HAVCR2/LAG3[low]. Of the two CD8+ T-cell clusters, T3 showed a terminally exhausted (Tex) profile (HAVCR2/LAG3[high]), while cluster T10, which represented only a minor fraction (i.e., 8%) of all non-malignant T-cells (Fig. 3J, K), met the criteria for a functional cytotoxic T lymphocyte (CTL) cluster with high expression of granzyme and perforin genes (Fig. 3L). The DN T-cell cluster T0 exhibited a Tpex profile, and the DN cluster T7 that of naive-like T-cells. Finally, T11, the smallest of the five T-cell clusters at 5%, was identified as a CD3-/CD4+ regulatory T-cell (Treg) cluster based on the expression of markers such as FOXP3. A similar phenotypic mapping of the Myeloid subset revealed that almost all clusters exhibited strong immunosuppressive features, including CD14[high], an M2-like profile, a C1Q+ profile of tumor-associated macrophages (TAM) that has been correlated with T-cell exhaustion[40]. Cluster M4 displayed an M-MDSC (mononuclear myeloid-derived suppressor cell) profile with high expression of marker genes such as S100A8/9[41] (Fig. 3M). In addition to these immunosuppressive features, we also found evidence of partial transdifferentiation, namely macrophage-to-myofibroblast transition (MMT), in a subset of M3/M0 cells. Myofibroblasts are a heterogeneous cell population that may arise from inflammatory macrophages

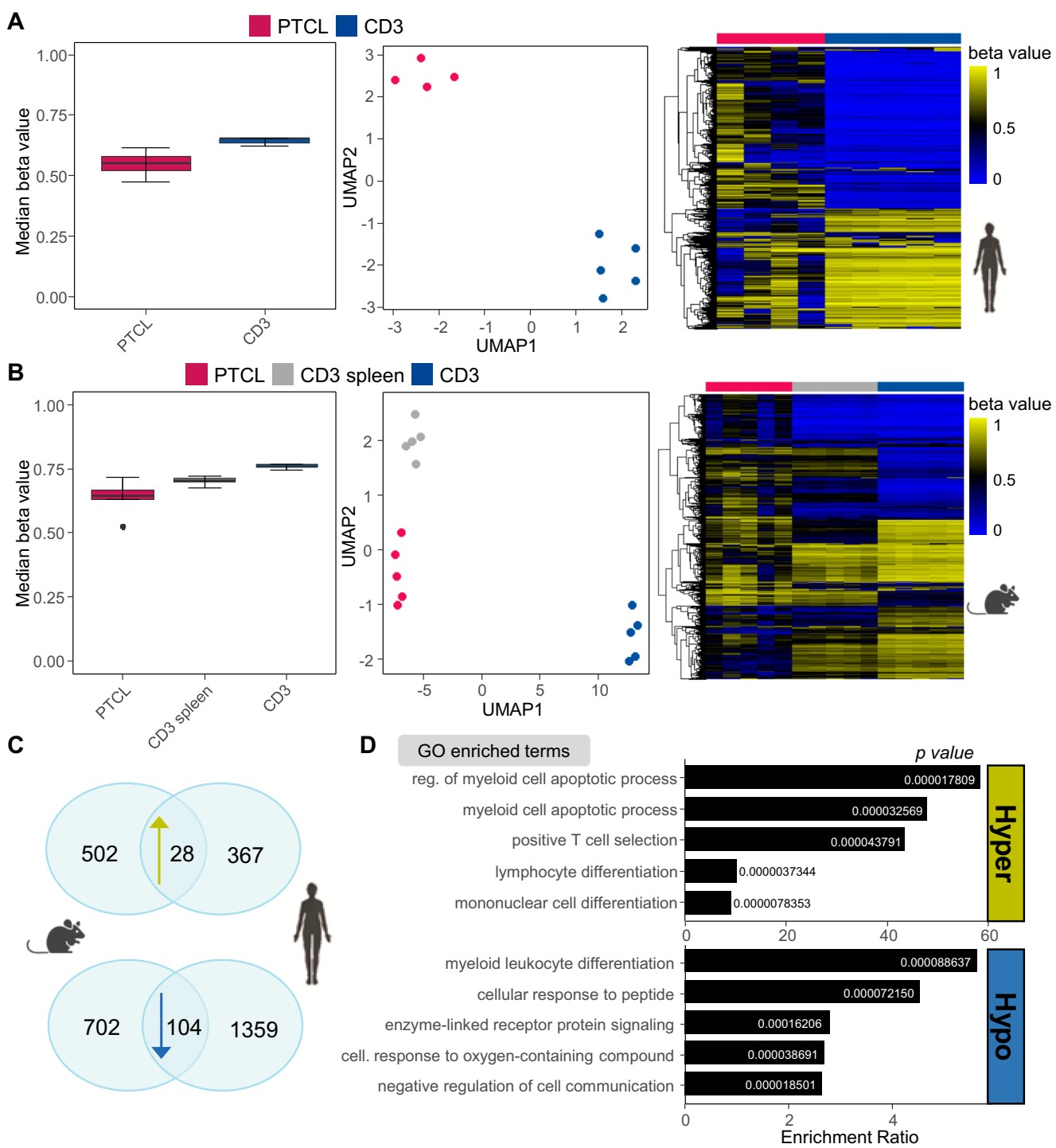

**Fig. 2 | Human and murine SMARCB1-deficient PTCLs share common methylation profiles. A** Median global DNA methylation in human PTCLs ($n = 4$) and CD3 T cells ($n = 5$). UMAP analysis based on the 10,000 most variable CpGs and 5 neighbors. Heatmap showing 10,000 most variable CpGs. Boxplot settings: middle, median; lower hinge, 25% quantile; upper hinge, 75% quantile; upper/lower whisker, largest/smallest observation less/greater than or equal to upper/lower hinge $\pm 1.5 \times$ IQR. **B** Median global DNA methylation in murine PTCLs ($n = 5$), splenic cells ($n = 5$) and Cd3 T cells ($n = 5$). UMAP analysis based on the 10,000 most variable CpGs and 5 neighbors. Heatmap showing 10,000 most variable CpGs. Boxplot settings: middle, median; lower hinge, 25% quantile; upper hinge, 75% quantile; upper/lower whisker, largest/smallest observation less/greater than or equal to

upper/lower hinge $\pm 1.5 \times$ IQR. **C** Overlap of genes hyper- or hypomethylated in murine in human PTCLs compared to Cd3 T cells. Hypomethylated cutoff: $\sigma/\sigma$max $> 0.4$, $q < 0.01$, Hypermethylated cutoff: $\sigma/\sigma$max $> 0.4$, $q < 1e\text{-}5$. **D** Biological process-associated GO terms assigned to concordantly hyper- and hypomethylated genes in PTCLs compared to Cd3 T cells. Over-representation analysis was performed using WebGestalt (https://2024.webgestalt.org/) with adjustment for multiple testing (Benjamini−Hochberg) (Table S9). Only gene sets with more than 10 genes were considered. (x axis, Enrichment ratio). Source data of **A**−**D** are provided as a Source Data file. **A**−**C** Created with BioRender.com released under a Creative Commons Attribution-NonCommercial-NoDerivs 4.0 International license.

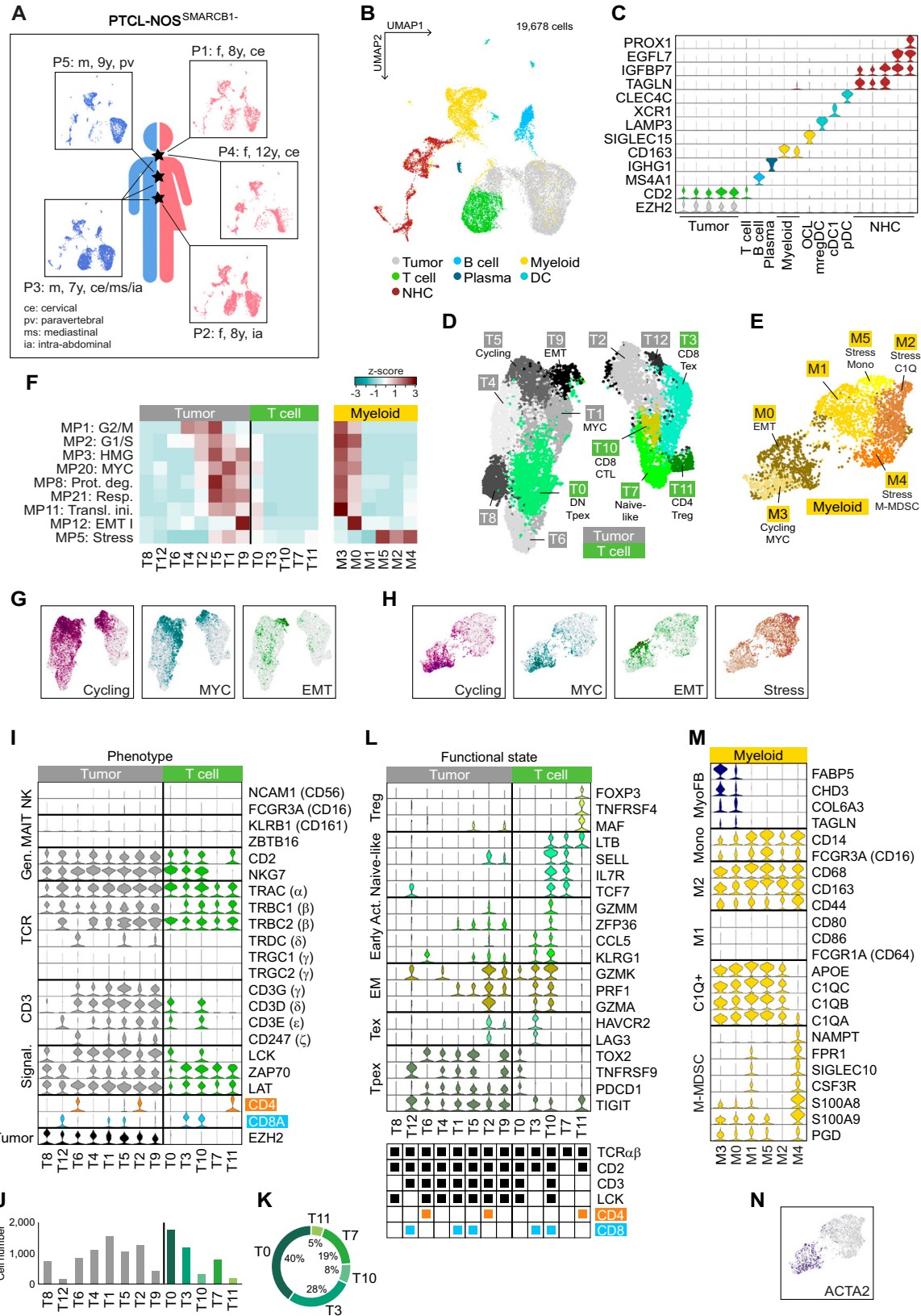

through an EMT-related process[42]. These cells showed co-expression of macrophage and myofibroblast markers such as CD68 and alpha-smooth muscle actin ACTA2 (Fig. 3N).

Based on the observation of shared regulatory programs in developmentally distinct cell populations, we sought to further eluci-date possible underlying communication pathways. We utilized CellPhoneDB[43] in conjunction with InterCellar[44] to probe potential cell-cell interactions (CCIs) between different cell populations. According to the total number of CCIs, three main interaction hubs emerged in this analysis, namely the myeloid TAM compartment, the tumor compartment, and the stromal NHC compartment (Fig. 4A). It was possible to crystallize several cancer-relevant communication paths

**Fig. 3 | Single-cell landscape of human PTCL-NOS^SMARCB1-. A** Overview of the five patient samples P1-P5 with information on gender, age, and tumor location. **B** Uniform manifold approximation and projection (UMAP) plot of the integrated scRNA-seq dataset. **C** Violin plot showing expression of cell type-specific marker genes in individual clusters. OCL osteoclast, DC dendritic cell, pDC plasmacytoid DC, cDC1 conventional type 1 DC, mregDC mature DC enriched in immunoregulatory molecules, NHC non-hematopoietic cell. **D** UMAP plot of the Tumor/T-cell subset and (**E**) the Myeloid subset after re-clustering. **F** Heatmap showing overlaps of cluster-specific DEG sets with signatures of cancer hallmark metaprograms[38]. Z-scores were calculated on a row-by-row basis. **G** Averaged expression levels of the identified gene signatures of tumor cell clusters T5 (Cycling), T1 (MYC) and T9 (EMT), and (**H**) of myeloid clusters M3 (Cycling and MYC), M0 (EMT) and M2/4/5 (Stress). **I** Classification of tumor and T-cell clusters.

NK natural killer cell, MAIT mucosal-associated invariant T-cells, Gen. generic T-cell marker, TCR T-cell receptor, Signal. TCR signaling. **J** Cell numbers of the individual tumor clusters (gray bars) or T-cell clusters (green bars), and **K** relative proportions of T-cell clusters as a circular diagram. **L** Different functional states of the tumor and T-cell clusters based on marker gene expression. Treg regulatory T-cell, Early Act. early activation state, EM effector memory T-cell, Tex terminally exhausted state, Tpex precursor-exhausted state. **M** Immunosuppressive features within the Myeloid subset. MyoFB myofibroblasts, Mono monocytes, M1/2 M1/2 polarization, M-MDSC mononuclear myeloid-derived suppressor cells. **N** ACTA2 expression in the Myeloid subset. Source data of **J** and **K** are provided as a Source Data file. **A** Created with BioRender.com released under a Creative Commons Attribution-NonCommercial-NoDerivs 4.0 International license.

from a large number of significant ligand-receptor (L-R) pairings. Of particular interest here are several chemokine signaling axes that are involved in TME remodeling through processes such as EMT and immunosuppression[45]. These include the CXCL12-CXCR4 axis with stromal NESC and FRC clusters and the CCL19-CCR7 and CXCL9-CXCR3 axes with FRC as senders; for all three signaling axes, different tumor/myeloid clusters such as T9/M0_EMT or T5/M3_Cycling represent the receiver cells (Fig. 4B–F). Activation of CXCL12-CXCR4 and CXCL9-CXCR3 promotes EMT and the mobilization of cancer cells into the pre-metastatic niche[46,47] and is linked to immunosuppression and T-cell exhaustion[48,49]. Other signaling axes include NESC and PC clusters as sender of the extracellular matrix (ECM) protein fibronectin (FN1) to the integrin receptor complexes ITGA4/B1 and ITGA5/B1 on several tumor and myeloid receiver clusters as well as BEC and LEC clusters as sender of the adhesion molecule PECAM1 to CD38+ tumor and myeloid cell populations (Fig. 4G, H). Fibronectin can induce EMT in human cancer cells[50], and PECAM1-CD38 signaling is involved in the formation of an immunosuppressive TME[51]. Finally, we identified two interaction axes between clusters of the same cellular compartment: first, the L-R pairing CD70-CD27 in the tumor cell compartment [both in the autocrine mode of the two clusters T5_Cycling and T1_MYC (T5::T5 and T1::T1) and in a paracrine mode with the receiver clusters T4, T6 and T9_EMT], secondly, the COL6A2-ITGA1/B1 pairing between different sender and receiver clusters in the NHC compartment (Fig. 4B, I). In hematological malignancies, co-expression of CD70 and CD27 has been shown to promote tumor stemness and proliferation[52], while collagens and integrins are closely linked to EMT and functionally involved in the metastasis of tumor cells[53].

## A mouse model recapitulates essential features of human PTCL-NOS^SMARCB1-

Since we were unable to include matching controls in our scRNA-seq analysis of the five human FFPE samples, we resorted to the PTCL mouse model to gain insights into the transformation process and the differences between tumor and healthy cells. For this purpose, the spleens of two tumor-bearing mice were isolated, and single cells were processed using the 10x Chromium platform. The sequence data obtained were merged with publicly available scRNA-seq data from two control spleens of healthy mice[54]. In this integrated object, 14,588 single cells were grouped into 24 different clusters, which were then annotated as described above. (Fig. 5A; Suppl. Fig. 9, Supplementary Data S8, 9). A comparison of mouse and human tumors revealed similar proportions of the various cell compartments, e.g., lymphoid populations such as non-malignant T- and B-cells (Fig. 5B). Especially in the latter, however, a large difference to healthy mouse spleens became apparent, in which B-cells accounted for 70% of the total cell number, while this number dropped to 10% and 8% in mouse and human tumors, respectively. At the same time, a significant increase in tumor-infiltrating myeloid cell populations was observed, from 5% in healthy spleens to over 30% and 20% in the mouse and human tumors, respectively. This inverse correlation between the number of B-cells

and myeloid cells when comparing WT to PTCL samples is further illustrated by the pie charts in Fig. 5B. We were able to confirm this phenomenon by multiplex immunofluorescence in murine spleen samples. Through a combination of specific antibodies for the tumor cell marker Ezh2, the pan B-cell marker B220/CD45R, and the neutrophil marker Ly6g, a significant decrease of splenic B-cells with a concomitant increase in Ly6g+ myeloid cell infiltration in tumors was revealed (Fig. 5C, D).

Differences between mouse and human tumors were observed at the level of specialized cell types. For example, in contrast to human tumors, which have a clearly monocyte/macrophage-dominated phenotype in the myeloid compartment, neutrophils form the largest myeloid lineage in the mouse. At the functional level, however, this difference is offset by the occurrence of similar (if operationally defined) cell types such as monocytic M-MDSC in human (Fig. 3M) and granulocytic PMN-MDSC in murine PTCL (Fig. 5A). The latter are found in cluster 9 and characterized by high expression of Ly6g, interleukin-1 beta (Il1b) and histidine decarboxylase (Hdc) (Suppl. Fig. 9A). Interestingly, Hdc+ PMN-MDSC have been linked to EMT and increased metastasis in murine tumor models[55]. The functional similarities also extend to CD14^high cells: in human PTCL they can be found in M-MDSC cluster M4 (Fig. 3M), in murine tumors in PMN-MDSC cluster 9 (Fig. 5G). Cd14^high cells are markers for an immunosuppressive TME and associated with tumor progression in the mouse spleen[41].

As with the human tumors, we also carried out a comparison of upregulated DEGs of each mouse cluster with cancer metaprograms (Fig. 5E, Supplementary Data S10, 11). This analysis revealed clear similarities with the functional tumor programming in patients. Significant matches with MPs were mostly found in tumor cell or myeloid clusters, such as a singular cell cycle cluster in the tumor and myeloid compartment (cluster 3 and cluster 20, respectively) and multiple EMT, MYC and Stress clusters, the latter being limited to the myeloid compartment as in human tumors (Fig. 5E, F). Furthermore, we found highly similar functional remodeling of the immune cell landscape in human and mouse tumors, characterized in both species by the exhaustion of T-/NK cells and by the infiltration of immunosuppressive myeloid cells. This is illustrated by the violin plot analysis in Fig. 5G, which shows that specific marker genes for T-/NK cell exhaustion and myeloid immunosuppression are selectively expressed in cells of the murine PTCL TME.

Analysis of the cell-cell interactions between tumor and infiltrating immune cells in the murine tumors revealed a consistent picture: the immune landscape, just as in patients, turns out to be highly immunosuppressive, proinflammatory and proangiogenic (Suppl. Fig. 10, Supplementary Data S12). As with human PTCL-NOS^SMARCB-, tumor-TME CCIs in the mouse are also strongly associated with EMT-related processes such as cell-matrix adhesion, cell migration and ECM organization (Suppl. Fig. 10B), in part via the same signaling axes (e.g., Cxcl9, Pecam1) (Suppl. Fig. 10C).

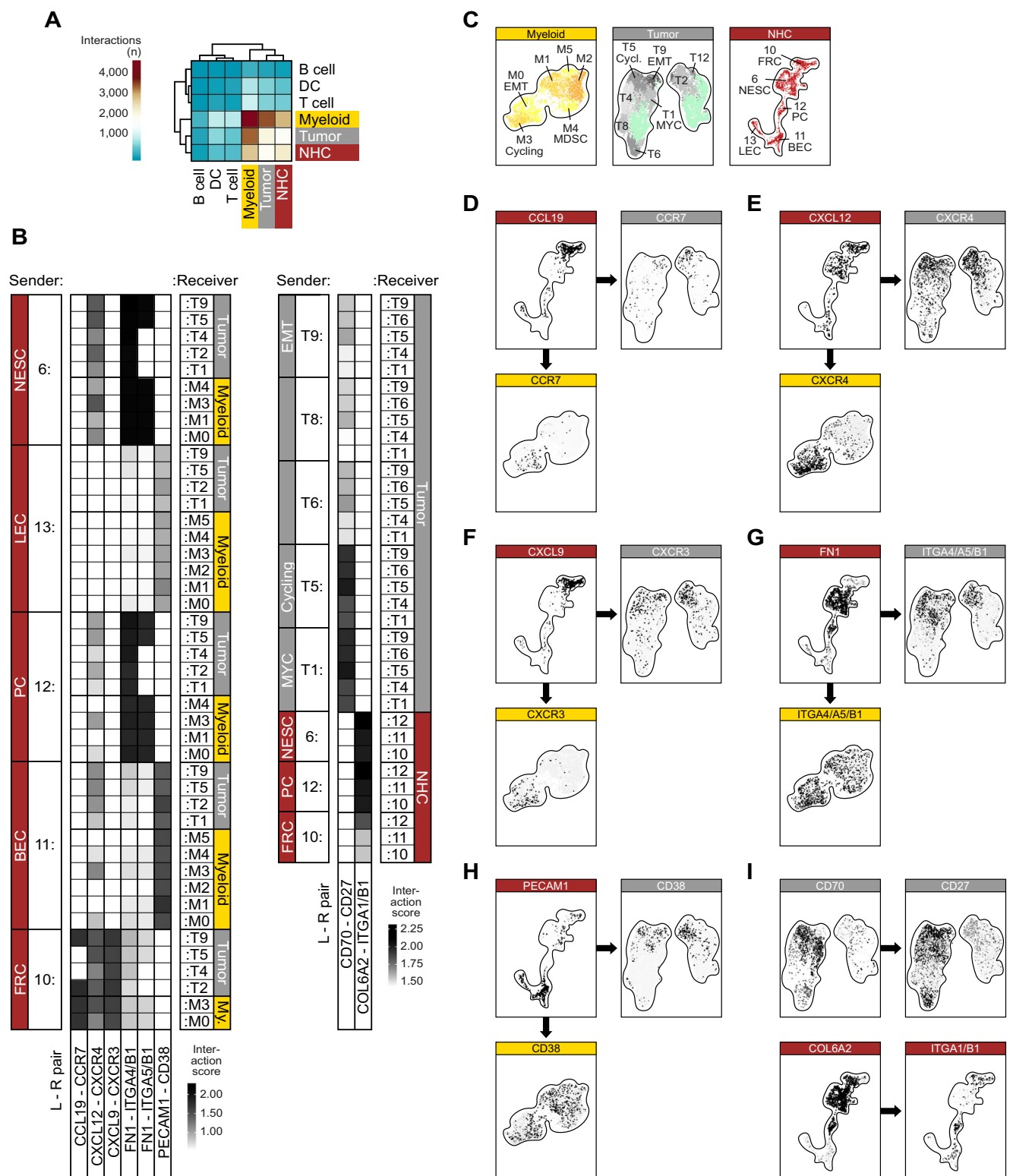

**Fig. 4 | Cell-cell communication in the tumor niche. A** Heatmap showing the number of significant ligand-receptor pairs for each cellular compartment. **B** Heatmap showing selected ligand-receptor (L-R) pairs for interactions of sender clusters (left) and receiver clusters (right). The greyscale indicates interaction score values. **C** Overview maps of the participating clusters within the three analyzed compartments. **D–I** Feature plots showing the average expression values of the displayed L-R pairs in the respective compartments.

## SAHA treatment mimics Smarcb1 re-expression in an in vitro model of PTCL-NOS*Smarcb1−*

The above observations suggest that epigenetic mechanisms contribute significantly to the development and progression of SMARCB1-negative PTCL-NOS. This motivated us to conduct preclinical experiments that could uncover possible therapeutic targets. We performed a drug screen with 140 epigenetically active compounds using the Smarcb1-negative murine PTCL cell line T15 as an in vitro model

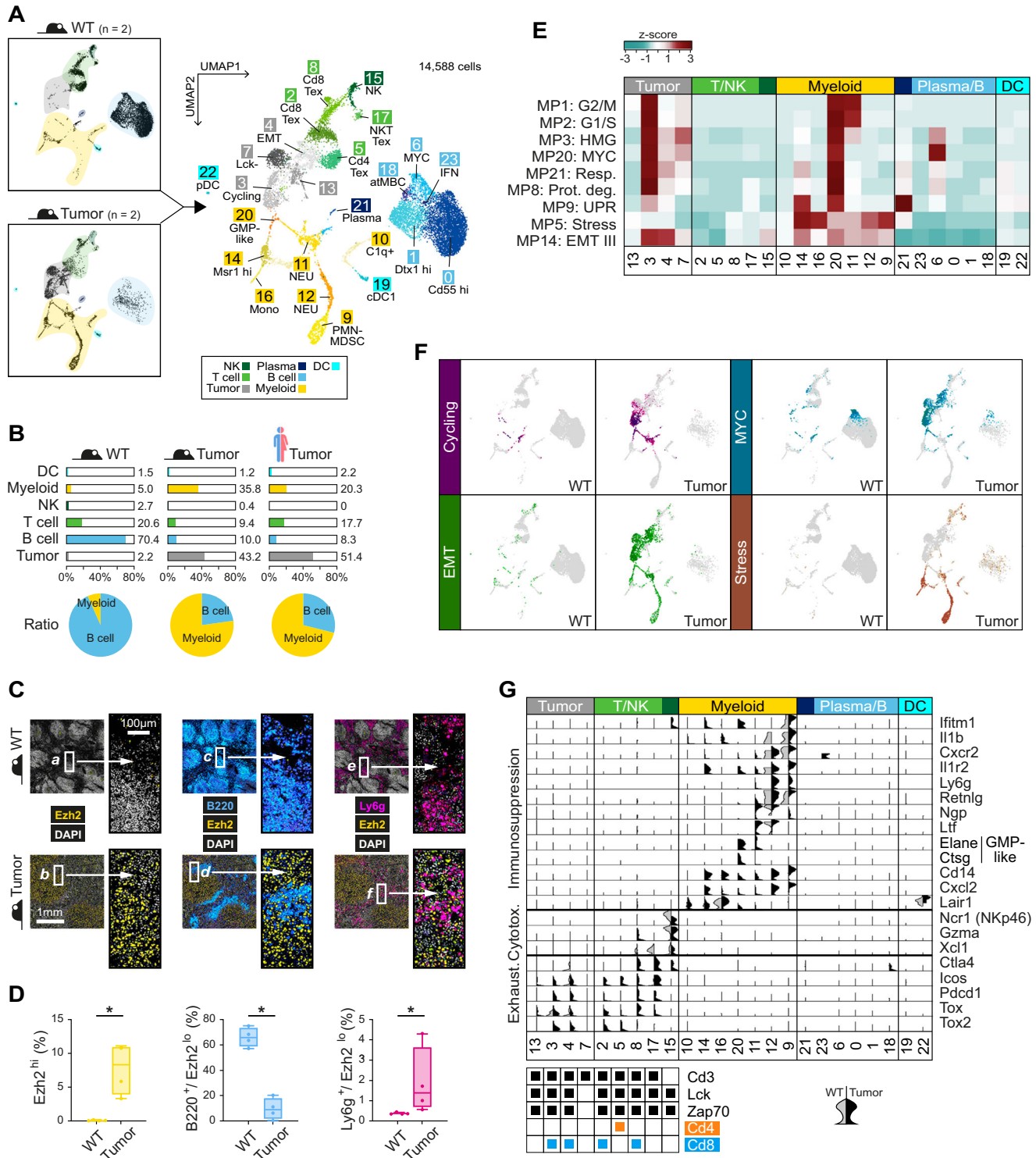

(Supplementary Data S13). The highest number of active compounds was found in the histone deacetylase inhibitor (HDACi) group (Suppl. Fig. 11A). This group also showed high efficacy in murine Smarcb1-negative PTCL compared to other non-Hodgkin lymphoma (NHL) cell lines (Fig. 6A). For further experiments, we selected SAHA (suberoylanilide hydroxamic acid/vorinostat) because it is FDA-approved and has been shown to be an effective agent in Smarcb1-negative rhabdoid tumors in our previous studies[56]. While SAHA treatment of the Smarcb1-negative PTCL cell line T15 did not induce significant cell cycle effects in the submicromolar range, at higher concentrations

(1 μM and 5 μM) it resulted in a strong induction of apoptosis with well over 90% dead cells (Fig. 6B, C; Supplement Fig. 11B, C).

In the next step, we planned to further investigate the effects of SAHA treatment of T15 cells on a global level using RNA sequencing (RNA-seq). At the same time, we also wanted to find out which genes are epigenetically silenced by the loss of Smarcb1. To this end, we modified the T15 cell line model by introducing a Smarcb1 conditional re-expression system (Smarcb1-RE) (Fig. 6D, E). After re-expression, a significant reduction in cell growth was observed (Fig. 6F), while cell viability and cell cycle were not affected.

**Fig. 5 | Murine PTCL-NOS^Smarcb1−^ recapitulates key features of human tumors.**
**A** UMAP plot showing 24 clusters of the integrated scRNA-seq dataset from two control spleen samples (WT) and two PTCL-NOS^Smarcb1−^ tumor samples. **B** Relative abundance of different cell types in murine WT spleens (left), PTCL spleens (middle), and human tumors (right; NB: in order to ensure comparability, the stromal cells were removed before quantification). The pie charts in the lower part show the ratio between B-cells and myeloid cells. **C** Multiplex immunofluorescence (IF) images of FFPE sections of murine PTCL-NOS^Smarcb1−^ and control spleen samples (WT: upper panels; tumor: lower panels). For better visualization, the white boxed areas (*a* to *f*) are enlarged (2.5x; scale bar = 100 μm). DAPI (gray) provides a nuclear counterstain, Ezh2 (yellow) defines malignant cells (Ezh2^hi^), B220 (blue) is used as a pan B-cell marker (B220^+^), and Ly6g (pink) as a marker for neutrophils (Ly6g^+^). **D** Quantitative analysis of IF images from (**C**). Four representative regions of interest (ROIs; size: 1500 × 1500 μm) were selected and analyzed for mouse WT and

Tumor samples. A Wilcoxon-Mann-Whitney test was calculated to determine if there are differences between WT and Tumor samples for all comparisons (*$p$ = 0.0286). Boxplot settings: middle, median; lower hinge, 25% quantile; upper hinge, 75% quantile; upper/lower whisker, largest/smallest observation less/greater than or equal to upper/lower hinge ±1.5 * IQR. **E** The heatmap shows the overlap between cluster-specific DEG lists and the cancer hallmark metaprograms. **F** Signature plots of the programs Cycling, MYC, EMT and Stress in cells from WT (left) and tumor (right) samples. **G** A split violin plot (left/gray half: WT; right/black half: tumor) illustrates the increase in T-cell exhaustion features (Exhaust.) with a simultaneous decrease in NK cytotoxicity (Cytotox.) markers (e.g., Ncr1/NKp46) as well as infiltration of immunosuppressive myeloid cells in tumor versus WT samples. Source data of **B** and **D** are provided as a Source Data file. **B** Created with BioRender.com released under a Creative Commons Attribution-NonCommercial-NoDerivs 4.0 International license.

We performed RNA-seq with three replicates, each of untreated T15 control cells, of SAHA-treated T15 cells, and of Smarcb1-RE cells (Fig. 6G). Subsequent bioinformatic analysis revealed a high level of agreement regarding enriched gene ontology (GO) terms in both treatment groups and included gene sets that are associated with (nervous system) developmental processes, but also gene sets that are functionally involved in cell growth, adhesion, motility, and cell-cell communication (Fig. 6H). More specifically, an overlap of 607 upregulated genes involved in cell growth and development is found in SAHA-treated and Smarcb1-expressing T15 cells (Fig. 6 I, J). In addition, the comparison between SAHA-treated and untreated T15 cells showed that HDACi leads to an upregulation of genes that are functionally involved in the differentiation of myeloid cells (Fig. 6K). This implies a reversal of the epi-phenotype of murine and human PTCL tumors, where we previously observed that genes particularly affected by DNA hypomethylation also include those of myeloid differentiation (cf. Fig. 2D). Taken together, SAHA treatment mimics the transcriptional changes seen by reintroducing Smarcb1 expression in T15 cells.

### SAHA treatment leads to remodeling of the immunosuppressive TME and reversal of the exhaustion phenotype

To investigate the effect of SAHA treatment in vivo, tumor-bearing mice were treated with SAHA for three weeks, then their spleens were isolated and analyzed with scRNA-seq. These sequence data (SAHA hereafter) were combined with those from untreated tumors (PTCL) and control spleens (WT) and then evaluated bioinformatically (Supplementary Data S14). Cell distribution and sample composition are shown in Fig. 7A. SAHA resulted in a moderate decrease in tumor cells and myeloid infiltration compared to untreated PTCL, while reversing the loss of B-cells (at least partially) and of non-malignant T-/NK cells (almost completely). When the number of B-cells is related to the number of myeloid cells, the effect of SAHA treatment, namely the replenishment of B-cells while suppressing myeloid infiltration, becomes very clear (see pie charts in Fig. 7A).

Next, we took a closer look at the quantitative and qualitative changes in the B-cell and T/NK cell compartments. Regarding B-cell subtypes, SAHA treatment elicited the replenishment of the B-cell compartment as it restored the pool of mature B-cells in the mouse spleen (Suppl. Fig. 12). In addition, the appearance of a progenitor B-cell-like population was observed, accounting for almost one-third of the total B-cell population in SAHA-treated animals (Suppl. Fig. 12C). Another cell population, termed PTCL B-cells because it originated in the tumor and was different from all other populations in healthy samples, almost completely disappeared from the spleens of SAHA-treated mice.

Analysis of T-cell subtypes in SAHA-treated versus untreated PTCL revealed (i) a relative increase in the proportion of Cd8+ effector T-cells, (ii) of Cd4+ and Cd8+ naive T-cells, (iii) of NKT and NK cells, and (iv) an almost complete extinction of terminally exhausted T-cells in

SAHA (Fig. 7B–D). A more detailed comparison of the expression of canonical marker genes in WT compared to PTCL and SAHA clearly shows the re-emergence of a functional T-/NK cell compartment in the latter, namely the recovery of cytotoxic properties with almost disappearing exhaustion features (Fig. 7E). Finally, we derived pseudo-time trajectories for both groups to correlate the contrasting activity states of Cd8+ effector cells in PTCL versus SAHA with potentially divergent differentiation pathways (Fig. 7F). While PTCL T-cells follow a pseudo-developmental trajectory from the naive to the terminally exhausted state, in SAHA this trajectory ends in the state of cytotoxic effector cells. Taken together, these data underscore the ability of SAHA to restore the functionality of key effector components of the adaptive and innate immune system in Smarcb1-negative PTCL-NOS.

## Discussion

PTCL-NOS is a rare, aggressive, and highly heterogeneous tumor entity[6,57,58]. Its clinical outcome with standard antiproliferative chemotherapies is currently unsatisfactory[59]. A deeper molecular characterization of this entity is required for the development of more efficient therapies.

Here we describe SMARCB1-negative PTCL-NOS as a potential molecular subtype of PTCL with relatively higher occurrence in children and young adults versus older adults. In parallel to this study, there were three SMARCB1-deficient cases described in children[60], further underlining the clear age association of this subtype. While the molecular origin of the loss of SMARCB1 in the human PTCL might be heterogeneous, including single nucleotide and structural genomic variants as well as probably epigenomic changes, we were able to model the disease phenotype in a targeted mouse model by inactivating Smarcb1 in mature T-cells. A strong concordance between naturally occurring SMARCB1-deficient PTCL in humans and in the targeted mouse model was found with regard to both the extent and direction of DNA methylation changes. Remarkably, the tumors of both species showed enrichment of DNA hypermethylation linked to genes involved in T-cell function and of DNA hypomethylation in genes linked to myeloid cell differentiation. The increase of myeloid cell populations was confirmed in the human and murine tumors by scRNA-seq analyses. Considering that the development of PTCL has been intimately linked to features of clonal hematopoiesis (CHIP) in both species[61], it is intriguing to speculate that SMARCB1 plays a pathogenic role in differentiation processes in early hematopoietic cells leading to PTCL.

By analyzing SMARCB1 RNA and protein expression levels in multiple subtypes of mature T-cell lymphomas, we could exclude T-PLL, MEITL, EATL, MF, AITL and ALK-negative ALCL being the human counterpart of the phenotype observed in Smarcb1-deficient mice.

While the tumor-suppressor function of SMARCB1 is well explained by its role in regulating chromatin accessibility, enhancer binding and differentiation[22,24,62], the role of the TME in the

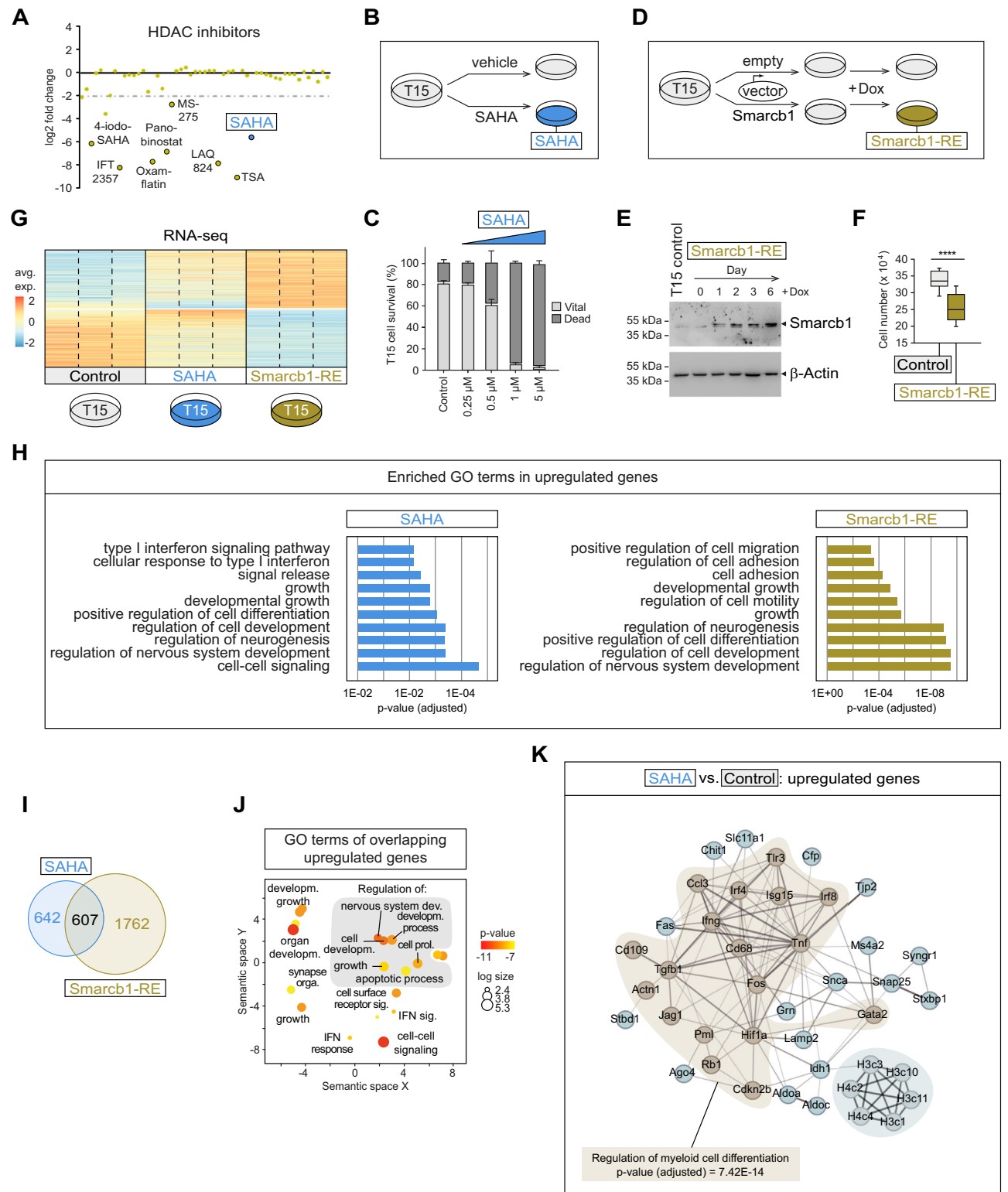

progression of SMARCB1-negative tumors is far less explored. In lymphomas and other cancers, it has emerged that the TME plays a decisive role in the pathogenesis and response to therapy[63–65]. In PTCL-NOS patients, specific immune cell signatures were found to be associated with superior clinical outcome[66]. We therefore paid particular attention to the TME and its interaction with the tumor cells in our study.

One central observation of our single-cell analyses of human and murine SMARCB1-negative PTCL was the extensive network of tumor-TME interactions. Yet, there are species-specific differences in the

detailed cellular architecture of the TME. First, we note the absence of stromal cells in mouse tumors, and second, the myeloid compartment is dominated by monocytes/macrophages in humans and by neutrophils in mice. This could have several possible reasons, including different tumor locations (spleen versus lymph nodes), different immune cell responses in mice compared to patients[67], or different isolation and/or processing procedures (fresh tumor tissue versus archival FFPE material). Despite this, there are many similarities at the functional level. The first striking feature is the adaptation of similar

**Fig. 6 | SAHA treatment recapitulates *Smarcb1* re-expression in PTCL-NOS[Smarcb1-].** **A** Effect of HDAC inhibitors on the viability of T15 cells versus seven non-Hodgkin lymphoma (NHL) cell lines. Cells were treated twice with 1 μM inhibitor over the course of five days and measured using an MTT assay. T15 cell viability was set in relation to NHL cells and is shown as log$_2$ fold change. **B** Scheme of SAHA treatment. **C** Dosage-dependent cytotoxic effects of SAHA on T15 cells (*n* = 4 biological replicates; data are presented as means +/– SD). **D** Scheme of Smarcb1 re-expression (Smarcb1-RE). T15 cells were transduced with an empty control vector or a Smarcb1 expression vector and induced by doxycycline (Dox; 0.5 μg/μl). **E** Representative immunoblots showing Dox-induced Smarcb1 re-expression. Beta-actin serves as a loading control. **F** Effect of Smarcb1 re-expression on T15 cell growth. The boxplots show median (center line), first and third quartile (bounds) and minima/maxima (whiskers) of Dox-treated (0.5 μg/μl; 72 h) T15 control and Smarcb1-RE cells (*n* = 3 biological replicates; paired two-sided *T* test; ****$p$ = 2.17E-05). Boxplot settings: middle, median; lower hinge, 25% quantile; upper hinge, 75% quantile; upper/lower whisker, largest/smallest observation less/greater than or equal to upper/lower hinge ±1.5 × IQR. **G** RNA sequencing (RNA-seq) analysis of T15 control cells, SAHA-treated (1 μM, 72 h) cells or Dox-induced (0.5 μg/μl, 72 h) Smarcb1-RE cells (3 biological replicates each). The heatmap shows the averaged gene expression values (avg. exp.) of significantly up- and down-regulated genes. **H** ToppGene (https://toppgene.cchmc.org/) was used to determine significantly enriched gene ontology (GO) terms associated with upregulated genes in SAHA or Smarcb1-RE cells. Shown are *p* values adjusted for multiple testing (Benjamini–Hochberg). **I** Venn diagram showing the overlap of SAHA and Smarcb1-RE upregulated genes. **J** GO analysis of overlapping genes using REVIGO[98]. The dot plot shows cluster representatives based on semantic similarities, where dot color indicates ToppGene-derived *p* values and dot size the frequency of the GO term in the underlying database. **K** Functional gene network analysis using STRING, showing that SAHA treatment regulates genes involved in myeloid cell differentiation (*p* value adjusted for multiple testing using Benjamini–Hochberg). Source data of **A**, **C**, **E**, **F** and **H–K** are provided as a Source Data file.

transcriptional metaprograms (Cycling, MYC, EMT, Stress) in the respective tumor cell and TME compartments. Furthermore, in both species, identical and/or similar signaling axes appear to serve this remarkable alignment of gene expression (re)programming. We were also able to observe similar patterns when comparing the relative proportions of higher-level cell compartments in human and mouse tumors. One of these patterns is the inverse correlation of myeloid and lymphoid (in particular B-cell) infiltration in PTCL. Our data suggest an immunosuppressive landscape that is promoted by multiple interactions between tumor cells, myeloid cells and, in the case of human PTCL, stromal cells in lymph node-localized tumor samples. Main characteristics are diminished infiltration of T-cells and NK cells which at the same time have highly activated and exhausted phenotypes. This relationship has already been well described[62,63,68]. Immunosuppressive cells of myeloid origin such as M-MDSC and PMN-MDSC inhibit anti-tumor immune responses by impairing the activation and function of T- and NK cells[64,69,70]. Furthermore, neutrophils can build extracellular neutrophil traps (NETs) around the tumor, which prevent T-cells and NK cells from being recruited to the TME[65]. Overall, we observed clear signs of a chronically inflamed TME. It is known that such a chronic inflammatory condition contributes to the depletion of immune effector cells; furthermore, it promotes angiogenesis and facilitates metastasis[64]. In addition, continuous triggering of signaling axes like CD70-CD27 (Fig. 4I) can also reduce NK cell numbers through apoptosis induction[71].

Effective therapeutic targeting will have to address both the malignant clone and the pathological TME. We propose SAHA, a pan-HDACi, as a promising therapeutic agent against SMARCB1-negative PTCL-NOS. Efficacy of several HDACi including SAHA is described for various hematological neoplasms[72]. SAHA is FDA-approved and in clinical use for relapsed or refractory (R/R) cutaneous T-cell lymphoma (CTCL) with tolerable toxic effects[73]. Romidepsin, a selective HDAC1 and 2 inhibitor, and belinostat, a broad-spectrum HDACi, are FDA-approved for R/R PTCL[74]. Our previous studies have demonstrated SAHA as an effective agent in *Smarcb1*-negative rhabdoid tumors[56]. In this context, it is also noteworthy that SAHA was identified as a potent drug for inducing reversal of epithelial-to-mesenchymal transition[75], a process that emerged in our study as a prominent motif within the tumor-TME communication of PTCL. Additionally, several studies have shown that EMT in tumors is related to the number of immunosuppressive cells in their TME[76]. Here we found that SAHA treatment mimics the transcriptional effects of *Smarcb1* re-expression in the TME of PTCL. Zhang et al.[77] described that HDACi treatment of exhausted lymphocytes restores their cytotoxic functionality in vivo, which could in part explain our observations. Removal of inhibitory signals could improve the trafficking of fully functional T-cells into the TME, turning it from a "cold" into a "hot" state, as previously reported with epigenetic modifiers[65,78]. Immunotherapy with immune checkpoint inhibitors has emerged as a promising approach for the treatment of hematologic malignancies, however, patients frequently do not respond or they become resistant to the treatment[59,79]. Specifically, patients with relapsed/refractory PTCL and CTCL treated with single-agent immunotherapy presented a high overall response rate but a very short progression-free survival[80], highlighting limited single-agent efficacy[81,82]. The identification of promising partners for future combination therapies with immune checkpoint inhibitors is an area of active clinical investigation[79]. As there is evidence for the reversibility of CD8 + T-cell exhaustion after immune checkpoint blockade, checkpoint inhibitors might be a potential treatment option for these patients in future combinatorial clinical studies. Our results provide the rationale for further investigations of combination therapies, including SAHA in PTCL-NOS[Smarcb1-].

## Methods

### Ethical approval

This study complies with all relevant ethical regulations. The SMARCB1 expression analysis in T-PLL has been approved by the Institutional Ethical Review Board of the Medical Faculty of Ulm University (21/16 and 463/19 (02.13. 2020)), in PTCLs from the TENOMIC Consortium Biobank by the Comité de Protection des Personnes Ile de France 08-009, in MEITL/EATLs by the Commission nationale d'éthique de la recherche sur l'être humain (CER-VD, protocol 382/14). CAYA PTCL-NOS patients were registered into the NHL-BFM study center database after written informed consent of the legal guardians had been obtained (Ethikkommission der Ärztekammer Westfalen-Lippe und der Westfälischen Wilhelms Universität; file number: 2017-077-f-S). As this study describes a very rare disease, we included all patients and did not select for age or sex/gender in advance. Sex/gender of patients was determined based on self-report.

### SMARCB1 expression analysis

SMARCB1 gene expression levels of T-cell lymphoma and T-PLL samples were mined from the TENOMIC database (LYSA consortium[29–35]) and from Patil et al.[28]. Additionally, the human T-cell leukemia cell line SUP-T11 (DSMZ, #ACC605) was analyzed. SMARCB1 protein expression was investigated by immunohistochemistry (IHC) in 15 PTCL-NOS patients from the TENOMIC tissue bank, 39 MEITL (monomorphic epitheliotropic intestinal T-cell lymphoma)[83] and 15 EATL (enteropathy-associated T-cell lymphoma)[84] patients using anti- SMARCB1 antibody (BD Bioscience, Clone 25/BAF47, #612110; 1:400 dilution). For PTCL-NOS, SMARCB1 protein expression was assessed in 14 adult patients[85] and 12 CAYA patients[86]. Additional SMARCB1-deficient PTCL-NOS cases were included as extension cohort (Table S4, Suppl. Fig. 4). RNA expression of pediatric PTCL-NOS cases was analyzed using the HTG Transcriptome analysis (details in supplements). SMARCB1 protein expression was also analyzed in a cohort of MF patients.

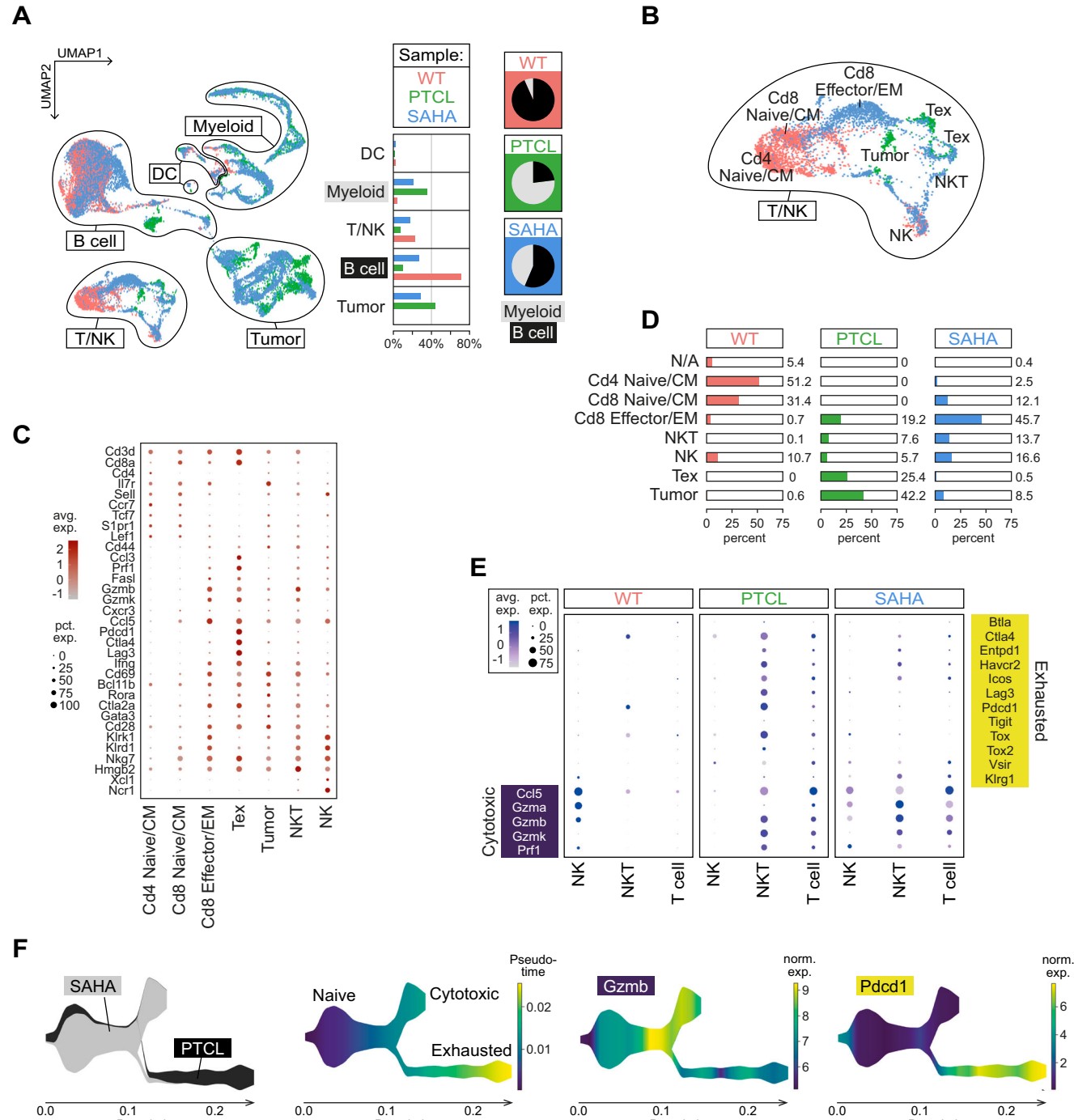

**Fig. 7 | SAHA treatment remodels the tumor microenvironment of PTCL-NOS$^{Smarcb1-}$ and reduces the exhaustion phenotype in vivo. A** Left: UMAP visualization of the integrated single-cell transcriptomes from control (WT), untreated, and SAHA-treated PTCL spleens (each $n = 2$ samples; 11,090 cells in total), Middle: relative contribution of different cellular compartments in the three sample groups; Right: pie charts showing the ratio between B-cells and myeloid cells in the three sample groups. **B** Expanded view of the T/NK cell compartment with a more detailed cell type annotation based on the dot plot shown in (**C**). **D** Proportions of distinct T/NK cell type subsets in the three sample groups. CM central memory T-cell, EM effector memory T-cell, Tex terminally exhausted T-cell, N/A unassigned cells. **E** Dot plot showing average expression levels (avg. exp.) and proportions of cells (pct. exp.) expressing exhaustion or cytotoxic marker genes in T-, NK and NKT cells from control, PTCL and SAHA-treated samples. **F** Trajectory analysis of Cd8+ T-cell clusters of untreated and SAHA-treated PTCL samples using STREAM[99]. Stream plot visualization of (from left to right): sample contribution (untreated = PTCL, black; treated = SAHA, gray), inferred phenotype and normalized expression (norm. exp.) of the cytotoxicity marker *Gzmb* and the exhaustion marker *Pdcd1* (PD-1) along the pseudotime axis. Source data of **D** are provided as a Source Data file.

## Mouse model and treatment

Protocols and animal housing were in accordance with all local regulatory authority guidelines (reference number TVA-84-02.04.2018.A296; State Government of North Rhine-Westphalia, Germany). Mice were housed under specific pathogen-free conditions in a 12-h light-dark cycle and received food and water *ad libitum*. *Cd4-cre::Smarcb1$^{fl/fl}$* animals were obtained by crossing *Cd4-cre*[87] (kindly provided by Dr. Maren Lindner, University Hospital Münster) and *Smarcb1$^{fl/fl}$* mice[27] (The Jackson Laboratory). In these mice, tumor formation typically occurs between week 9 and week 12; however, no specific maximum

tumor size is defined in this model. Lymphomas manifest by infiltration of the spleen and lymph nodes, and tumor formation is generally accompanied by deterioration of the general condition and palpable splenic enlargement. Mice were monitored daily using a score sheet for these characteristics and other clinical/behavioral symptoms such as apathy, reduced food/water intake, respiratory difficulties, or motor abnormalities. Experiments were terminated and animals euthanized if animals showed poor general condition or any of the above symptoms. For HDACi in vivo experiments, SAHA (Cayman Chemical Company, #10009929) was applied intraperitoneally three times per week for three weeks in a concentration of 50 mg/kg. See Table S7 for further details.

### DNA methylation analysis

Mouse spleens were dissected and treated with StemPro Accutase (Gibco, #A1110501), Cd3+ T-cells isolated by FACS and genomic DNA extracted using Quick-DNA Microprep Kit (Zymo Research, #D3020). DNA methylation profiles of primary human PTCL ($n = 4$) were generated using the Infinium MethylationEPIC BeadChip (Illumina). For comparison, we mined publicly available data of different T-cell subpopulations generated with the Infinium HumanMethylation450 BeadChip (Illumina)[88–95] (see Table S6). In addition, DNA methylation profiles of the PTCL-NOS$^{Smarcb1-}$ mice ($n = 5$) and corresponding control groups including non-neoplastic samples isolated from the spleen ($n = 5$) and splenic Cd3+ sorted cells ($n = 5$) were generated using a custom service for Infinium Mouse Methylation BeadChip.

### Mouse tissue analysis

Isolated murine spleens were fixed and H&E-stained according to standard protocols. IHC stains were performed on a Ventana Bench-Mark XT using the ultraView Universal DAB detection Kit (Roche, #760-500) with anti-SMARCB1 antibody (BD Bioscience, Clone 25/BAF47, #612110; 1:50 dilution). Images were captured with an Olympus BX43 microscope.

### Bulk RNA sequencing of T15 cells

RNA was isolated from cell pellets of T15 control cells, SAHA-treated (1 μM, 72 h) T15 cells, or Dox-induced (0.5 μg/μl, 72 h) Smarcb1-RE cells (3 biological replicates each) using the RNeasy Mini Kit (Qiagen, Hilden, Germany; #74104) according to the manufacturer's protocol. Quality, purity and concentrations of individual RNAs were determined using the 2100 Bioanalyzer instrument (Agilent Technologies, CA, USA). RNA-seq libraries were prepared using the NEBNext Ultra II Directional RNA Library Prep Kit (New England Biolabs, MA, USA; #E7765) according to the manufacturer's instructions. All libraries were sequenced as single-end reads using the NextSeq 500 sequencing platform (Illumina, CA, USA) with the NextSeq 500/550 reagent kit v2.5 at the Core Facility Genomics (CFG) of the University Hospital Münster (Münster, Germany).

### Single-cell RNA sequencing of murine and human tumor samples

Sample preparation for SMARCB1-negative murine and human PTCL-NOS samples for scRNA-seq is described in detail in the Suppl. Methods. The murine samples were processed using the Chromium Single-Cell 3' Gel Bead Kit v2 (10X Genomics, CA, USA) according to the manufacturer's protocol and sequenced by the CFG on the NextSeq 500 sequencing platform (high performance kit, 75 cycles, v2 chemistry). Human samples were processed using 10x Genomics' protocols for Chromium Next GEM Single-Cell Fixed RNA Profiling technology. Human samples were sequenced as dual-index libraries by CFG on Illumina's NextSeq 2000 and NovaSeq 6000 sequencing systems. See Suppl. Methods for further details.

### Drug screen

Cell lines used in the drug screen were murine Smarcb1-negative T15 cells (gift from Charles W. M. Roberts, Dana-Farber Cancer Institute, Boston, USA) and the human Non-Hodgkin lymphoma (NHL) cell lines Jurkat (T-ALL), Karpas-299, SR-786, SU-DHL-1 (all ALCL), Raji, Daudi (Burkitt lymphoma) and U-937 (histiocytic lymphoma). All cells were maintained at 37 °C and 5% CO2 and cultured as described in Supplementary Table 11. The epigenetic drug library (Cayman Chemical, #11076, lot #0522205) comprised 140 compounds (see Supplementary Data S13). Cells ($4 \times 10^3$ cells/50 μL) were used in five replicates at a final concentration of 1 μM. Cell viability was measured 120 hours after the first treatment using MTT assay[96]. Log$_2$FC values were determined for each cell line and the results from T15 cells were referenced to the mean of all other NHL cell lines to evaluate the relative efficacy of the drugs.

### Multiplex immunofluorescence

For multiplexed immunofluorescence analysis, slices of PTCL-NOS$^{Smarcb1-}$ and corresponding murine control spleens were stained in the MACSima imaging system using antibodies against B220 (RA3-6B2, Miltenyi Biotec, APC, 1:50), Ly6G (1A8, Miltenyi Biotec, PE, 1:50) and EZH2 (REA907, Miltenyi Biotec, APC, 1:50). For further details see Suppl. Methods.

### Smarcb1 re-expression

For re-expression in T15 cells, Smarcb1 cDNA was introduced in the plasmid pInducer20 (Addgene, MA, USA; Plasmid #44012). Lentiviruses were generated by co-transfection of Smarcb1-pInducer20[97] and the two packaging plasmids psPAX2 and VSV-G (Addgene; Plasmid #12260 and #8454) into the Lenti-X 293 T cell line (Takara Bio USA, Inc., #632180) using the transfection reagent transIT®-lenti (Mirus Bio, WI, USA; #6600). 48 hours after transfection, the supernatant was harvested and frozen. After thawing the virus supernatants, T15 cells ($1 \times 10^6$) were transduced for 8 hours with 750 μl of viral supernatant, 250 μl of fresh medium and freshly thawed Polybrene (10 μg/ml; Sigma-Aldrich; #TR-1003). The cells were then harvested, centrifuged twice (1200 rpm for 6 min), resuspended in fresh medium and seeded on a six-well plate. Antibiotic selection was carried out for at least 14 days with Geneticin/G418 sulfate (Thermo Fisher Scientific, #11811-023) and several independent, stable clones were established. Protein expression was determined by quantitative real-time qPCR and by Western blot analysis.

### Statistics and reproducibility

Methods used for statistical hypothesis testing and exact $n$ numbers are directly stated in the figure legends. In general, the significance level was set to 0.05. Where applicable, $p$ values were corrected for multiple testing using Benjamini–Hochberg. Boxplots were generated using the default ggplot2 geom_boxplot settings (middle, median; lower hinge, 25% quantile; upper hinge, 75% quantile; upper/lower whisker, largest/smallest observation less/greater than or equal to upper/lower hinge $\pm 1.5 \times$ IQR).

### Reporting summary

Further information on research design is available in the Nature Portfolio Reporting Summary linked to this article.

## Data availability

Full descriptions of experimental procedures and bioinformatic methods can be found in the Supplemental Methods. We used publicly available scRNA-seq data of mouse control spleens from the Tabula Muris Consortium available at figshare (https://figshare.com/articles/dataset/Single-cell_RNA-seq_data_from_microfluidic_emulsion_v2/5968960). Raw data generated in this study have been deposited on Gene Expression Omnibus (GEO) under accession numbers GSE190273

for bulk RNA-seq data of T15 cell lines, GSE190274 for mouse PTCL single-cell RNA-seq, GSE249566 for human and murine PTCL DNA methylation array data and GSE254299 for human PTCL single-nuclei RNA-seq data. Source data are provided with this paper.

## Code availability

No source code or custom scripts were developed in this study. Data analysis was performed using publicly available packages. Details are available upon request.

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

## Acknowledgements

We thank professor Charles W. Roberts (St. Jude, Memphis, TN) for providing T15 cells. We thank Annegret Rosemann and Elisabeth Jung (Department of Pediatric Hematology and Oncology, University Children's Hospital Münster) and the teams of the tumor genetic and epigenetic laboratories at University Medical Center Ulm, particularly Britt König and Nnamdi Okeke, for excellent technical assistance. K.K. is supported by funds from "Gesellschaft für KinderKrebsForschung", Deutsche Krebshilfe e.V. (111784), Deutsche Kinderkrebsstiftung (DKS 2018.06), and Innovative Forschung Münster (I-KE121502). RS received support from funds from Deutsche Krebshilfe, e.V. (70114040) and DFG (SFB1074, project B9). Part of the analyses were conducted in the framework of the EuroTCLym Study within TRANSCAN.2 ERA-NET (BMBF; grant 01KT1907) to P.G., L.D.L., and R.S. S.T.B. received support from the Kinderkrebshilfe Münster e.V., Innovative Forschung Münster, and the Young Investigator Fund of the Fritz Thyssen and Schering Foundation. O.S. received support from DFG (502158695) and DFG TRR332 Z1. Figures were partly created with BioRender.com.

## Author contributions

Conceptualization: A.F., T.K.A., N.M., M.I., S.T.B., R.S., K.K. Methodology and experiments: N.M., J.M., P.P., F.W.F., M.R., S.T.B., C.G., M.G., E.K., V.M. Data analysis: A.F., T.K.A., N.M., M.I., J.M., S.G., P.P., F.W.F., M.R., A.V., R.W., S.B., S.D., D.M., C.I., V.M., W.H., H.H., St.R., F.O., C.W., E.M., K.K. Patient cohorts and histopathological assessment: G.D.S., Ra.S., A.C., A.D., J.B., M.H., S.C., SiS., A.Fo., C.R., A.T., M.B.P., F.A.A., K.M., U.K., V.K., L.V., E.M., V.F., P.G., B.B., W.K., Ld.L. Original draft: A.F., T.K.A., N.M., M.I., R.S., K.K. Supervision: T.K.A., M.D., Sa.S., J.V., U.S., O.S., R.S., K.K. Funding acquisition: K.K.

## Funding

## Competing interests

The authors declare no competing interests.

## Additional information

Anja Fischer [1,28], Thomas K. Albert [2,28], Natalia Moreno[2,28], Marta Interlandi[2,3,28], Jana Mormann[2], Selina Glaser[1], Paurnima Patil[1], Flavia W. de Faria[2], Mathis Richter [4], Archana Verma[2], Sebastian T. Balbach [2], Rabea Wagener[1], Susanne Bens [1], Sonja Dahlum[1], Carolin Göbel [5,6], Daniel Münter [2], Clara Inserte[3], Monika Graf[2], Eva Kremer[2], Viktoria Melcher[2], Gioia Di Stefano[7], Raffaella Santi[7], Alexander Chan [8], Ahmet Dogan [8], Jonathan Bush [9], Martin Hasselblatt[10], Sylvia Cheng[11], Signe Spetalen[12,13], Alexander Fosså[14], Wolfgang Hartmann [15], Heidi Herbrüggen[2], Stella Robert[16], Florian Oyen[5], Martin Dugas [3,17], Carolin Walter[3], Sarah Sandmann [3], Julian Varghese [3], Claudia Rossig[2], Ulrich Schüller [5,6,18], Alexandar Tzankov [19], Martin B. Pedersen[20], Francesco A. d'Amore [20,21], Karin Mellgren [22], Udo Kontny[23], Venkatesh Kancherla[24], Luis Veloza[24], Edoardo Missiaglia[24], Virginie Fataccioli[25,26], Philippe Gaulard[26], Birgit Burkhardt [2], Oliver Soehnlein [4], Wolfram Klapper [27], Laurence de Leval [24], Reiner Siebert[1,29] & Kornelius Kerl [2,29] ✉

[1]Institute of Human Genetics, Ulm University Medical Center, Ulm, Germany. [2]Department of Pediatric Hematology and Oncology, University Children's Hospital Münster, Münster, Germany. [3]Institute of Medical Informatics, University of Münster, 48149 Münster, Germany. [4]Institute for Experimental Pathology, Center for Molecular Biology of Inflammation, University of Münster, Münster, Germany. [5]Department of Pediatric Hematology and Oncology, University Medical Center Hamburg, Eppendorf (UKE), 20251 Hamburg, Germany. [6]Research Institute Children's Cancer Center, 20251 Hamburg, Germany. [7]Pathological Anatomy Section, Careggi University Hospital, Florence, Italy. [8]Department of Pathology, Hematopathology Service, Memorial Sloan Kettering Cancer Center, New York City, NY, USA. [9]Division of Anatomical Pathology, British Columbia Children's Hospital and Women's Hospital and Health Center, Vancouver, BC, Canada. [10]Institute of Neuropathology, University Hospital Münster, 48149 Münster, Germany. [11]Division of Pediatric Hematology/Oncology/BMT, Department of Pediatrics, British Columbia Children's Hospital, University of British Columbia, Vancouver, BC, Canada. [12]Department of Pathology, Oslo University Hospital, Oslo, Norway. [13]Institute of Clinical Medicine, Faculty of Medicine, University of Oslo, Oslo, Norway. [14]Department of Oncology, Oslo University Hospital-Norwegian Radium Hospital, Oslo, Norway. [15]Division of Translational Pathology, Gerhard-Domagk-Institut für Pathologie, Universitätsklinikum Münster, Albert-Schweitzer-Campus 1, Gebäude D17, 48149 Münster, Germany. [16]Department of Medicine A, Hematology, Oncology, and Pneumology, University Hospital Münster, Münster, Germany. [17]Institute of Medical Informatics, Heidelberg University Hospital, Heidelberg, Germany. [18]Institute of Neuropathology, University Medical Center Hamburg-Eppendorf (UKE), 20251 Hamburg, Germany. [19]Institute of Medical Genetics and Pathology, University Hospital Basel, Basel, Switzerland. [20]Department of Hematology, Aarhus University Hospital, Aarhus, Denmark. [21]Department of Clinical Medicine, Aarhus University, Aarhus, Denmark. [22]Department of Pediatric Oncology and Hematology, Sahlgrenska University Hospital, The Queen Silvia Children's Hospital, Gothenburg, Sweden. [23]Section of Pediatric Hematology, Oncology, and Stem Cell Transplantation, Department of Pediatric and Adolescent Medicine, RWTH Aachen University Hospital, Aachen, Germany. [24]Institute of Pathology, Department of Laboratory Medicine and Pathology, Lausanne University Hospital, Lausanne, Switzerland. [25]INSERM U955, Université Paris-Est, Créteil, France. [26]Département de Pathologie, Hôpitaux Universitaires Henri Mondor, AP-HP, INSERM U955, Université Paris Est Créteil, Créteil, France. [27]Department of Pathology, Haematopathology Section and Lymph Node Registry, University Hospital Schleswig-Holstein, Kiel, Germany. [28]These authors contributed equally: Anja Fischer, Thomas K. Albert, Natalia Moreno, Marta Interlandi. [29]These authors jointly supervised this work: Reiner Siebert, Kornelius Kerl. ✉e-mail: Kornelius.kerl@ukmuenster.de

