## [Peer Review File · Nature Communications]

REVIEWER COMMENTS

Reviewer #1 (Remarks to the Author): Expert in lymphoma genomics, scRNAseq, and in vivo models

Moreno et. al. identified a subtype of peripheral T cell lymphoma (PTCL) that appears to be driven by loss of SMARCB1. Using human samples and a mouse model, they demonstrated that this resulted from DNA hypermethylation in malignant T cells. With scRNA-seq, they identified additional immunosuppression in the TME. Using an HDAC inhibitor identified in a viability drug screen, they could remodel this TME in the mouse model. The molecular mechanisms of PTCL development remain incompletely characterized and this study can potentially fill an important gap. However, I have several concerns outlined in detail below about methodology in the analysis and the conclusions reached. There are also multiple instances in the manuscript where results are inadequately explained/characterized. I would encourage the authors to walk the reader through each result carefully. A few examples of this are included below.

1. The authors claimed that 'SMARCB1-negative PTCL-NOS is a novel molecular subtype of PTCL enriched in young patients'. How does SMARCB1 loss correlate with currently described PTCL-NOS subtypes PTCL-GATA3 and PTCL-TBX21? What evidence supports that this is a novel subtype that is mutually exclusive with currently established subtypes?

2. Additionally, the number of samples examined is quite low to conclude that this subtype is associated with CAYA. In the discussion (line 416), the authors link PTCL with CHIP. Since CHIP is associated with increased age, how would this explain SMARCB1 loss as a driver in younger patients?

3. The scRNA-seq results are incompletely characterized.

- Please walk us through detailed cell type annotation, the choice of marker genes used (eg. how were MDSCs differentiated from macrophages?).

- 'The cell distribution and sample composition are shown in Figure 6A-C and Suppl. Figure 11' : This is an example where a description of the actual results would be highly beneficial to the reader.

- 'Tumor cells most commonly interacted with Myeloid prog., monocytes and macrophages (Suppl Figure 9C)': how was 'most' determined?

- What is the significance of the disappearance of tumor cluster 25 after SAHA treatment?

- What are the adjusted p-values for the genes identified in Fig. 7A?

- In previous analysis, the authors demonstrated DNA hypomethylation in genes linked to myeloid cell differentiation with Smarcb1 deficiency. What was the effect of SAHA on myeloid gene expression?

- It appears that scRNA-seq with control and SAHA treatment was conducted in n=2 mice per group. If the authors have access to fixed/frozen samples from the mouse cohort, proportion and TME reprogramming findings could be validated eg. by IHC in some more samples.

4. Drug screen: How were the 140 drugs chosen? How was SAHA chosen from these drugs? It would help the readers a lot if you could walk us through your results. 'After validating different HDACi by cell viability, apoptosis and cell cycle assays, we selected SAHA for further experiments (Suppl. Figure 10B,C,D)' – what do these results indicate compared to other drugs in the screen?

Minor:

1. In the RNA-seq analysis of Smarcb1 re-expression (relating to Fig. 5B): details are all in the legend.

Please mention in the text that you carried out RNA-seq. Are these results from only one experiment?

The authors mentioned that '...viability and cell cycle were unaffected (not shown)'. Please include data on negative results. If neither of these were affected, is the reduction in cell growth a result of

apoptosis? Could the authors explain and discuss this finding?

2. Please double check figure callout accuracy. Some of the instances with wrong callouts include - Line 317 “..by exhaustion markers such as Ctl4, Pdc1 or Tox (Suppl. Figure 7A)”, should be Supplementary Figure 9A. Line 350 (Suppl. Figure 10A) should be Supplemental Figure 10B, Line 355: Figure 9F, should be Supplementary Fig. 10F

3. Table S8: Tabula Muris links are broken. Please enter the actual numbers in the table. If these are unavailable, please describe the dataset in detail in the methods and how it was acquired and processed in the analysis.

4. There are instances of missing references. As an example: Line 330: ‘..T-cells are attracted to myeloid cells via Cxcl9/10, which then represses their function via..’

5. Supplemental methods: ‘Due to the FFPE tissue, quality of the DNA was below average.’ What were the metrics and is the read quality reliable for CNV analysis? How was this determined?

6. I might have missed this, but it does not seem like Supplemental Fig. 4B FISH results were described in the manuscript.

7. Fig. 2D: this needs to be corrected for FDR.

Reviewer #2 (Remarks to the Author): Expert in T-cell lymphomas and genetically-engineered mouse models

The manuscript entitled ‘Lack of SMARCB1 expression characterizes a subset of human and murine peripheral T-cell lymphoma (PTCL)’ by Moreno et al. identified a subpopulation of PTCL-NOS young patients that lack SMARCB1 that currently have very poor outcome. They had previously developed a PTCL-NOS smarcb deficient mouse model and here they focus on the differences in immune cell phenotype from healthy tissues compared to malignant tissue (Tumor cells and TME) in these mice. They compared both in terms of methylation profiles and found a very strong similarity in methylation profiles with corresponding PTCL-NOS smarcb deficient PATIENTS in T cells and myeloid cells. Extensive single cell RNA expression analysis of the murine tumors showed a correlation of this methylation pattern with immune suppression, exhaustion and inflammation in the TME. After identification of methylation interfering drugs (histone deacetylase inhibitors) in vitro that were effective in inhibiting corresponding PTCL-NOS smarcb deficient cell lines, they used one of them ‘SAHA’ in their preclinical mouse model. They showed again by single cell RNA expression data that the SAHA treatment completely reshaped the tumor microenvironment with CD8 T cells reverting from exhausted to effector cell phenotype, a stronger infiltration of NK and NKT and loss of tumor suppressive myeloid cells compared to untreated groups.

The reviewer agrees that these results argue for the use of pan-HDACi inhibitors in combination with other drugs for treatment of these specific subgroup of patients, which are in need of new therapeutic regimens.

I recommend this paper for publication but I have some comments though I would like to see addressed before

Comments to the authors

- 1) The preclinical mouse model used by the authors was previously developed (two publications) but it might be more convincing to give an extensive comparison between the PTCL-NOS SMARCB1- mouse tumor development and the counterpart patients in the introduction. Maybe a summarizing table comparing the immune phenotypes, pathology etc... in supplementary data might help a lot to convince the reader that these mice are valid as preclinical model for PTCL-NOS SMARCB1-patients.
- 2) Although the author show that PTCL-NOS patient are more frequently low in SMARCB1, quite some AITL patients also show lower expression; AITL though is a malignancy more touching older people; these AITL patient with low SMARCB1 are they younger than the overall AITL patient population and might they also develop a different methylation pattern and benefit from this treatment. Can the authors discuss this.
- 3) Related to the question 1, In figure 3G a lower level of B cells is detected in the PTCL spleens compared to controls spleens while myeloid populations are more prominent. Is this also true for human PTCL-NOS? the authors should show these data or mention is this is already previously published.
- 4) Figure 5. It might have been interesting to compare the differentially expressed genes not only in smarcb1 re-expressing cells versus untreated and SAHA treated but also cells without smarcb1 Knock-out to evaluate in how far SAHA treatment is reversing compared to a PTCL cell line with normal smarcb1 expression levels?
- 5) In Figure 6E, how do the authors explain the complete disappearance of the high number of B-cells in the tumor upon SAHA treatment.
- 6) As performed in Figure 4, can the authors also indicate the changes in crosstalk in terms of cytokines and chemokines which can explain the reduction of infiltrating myeloid cells and augmentation of T and NK more precisely in the TME?
- 7) Ethical statements on using patient data and performing mouse experiments should be indicated in the material and methods

Minor

- 1) Figure 3E Mistake on the axis 'rogarnisation' should be reorganization
- 2) Figure 4B Although this representation is clear the singly or double arrows are not clear and should be enlarged.

Reviewer #3 (Remarks to the Author): Expert in lymphomas, genetically-engineered mouse models, and therapy

In this manuscript by Moreno and colleagues, a subgroup of PTCL with loss of SMARCB1 expression was identified, mostly affecting younger patients. Epigenomic and transcriptional sequencing in human PTCL-SMARCB1- lymphomas and in T-cell lymphomas developed in a mouse model with conditional deletion

of Smarcb1 in Cd4 T cells, revealed similarities in mouse and human samples related to common pathways of lymphomagenesis. Analysis of the TME inferred from scRNAseq in murine spleen tumors revealed cellular and functional changes involving the lymphoma-microenvironment interaction network. Following a limited drug screen, HDAC inhibitors including SAHA showed increased anti-lymphoma activity against human T-cell lymphoma cells, and were able to remodel the cellular immune microenvironment in mouse T-cell lymphomas. The authors therefore claim that SHA could have a particular therapeutic value in PTCL with loss of SMARCB1.

Overall, the manuscript combines transcriptional and epigenomic sequencing studies in human samples and in a mouse T-cell lymphoma model driven by Smarcb1 loss. This is well written and provides new information, with potential clinical/therapeutic interest. I am however suggesting additional studies to be considered, which could strength the conclusions and improve the final quality of this work.

Major concerns

The number of T-cell lymphomas samples analyzed in Figure 1 is limited. Regarding Figure 1A, expression of PTCL-NOS is relatively lower than in AITL, but similar to NK/TCL and HSTL. These data are not clear to me, in part because I do not find in the figure legends the definition of the acronyms for the T-cell lymphoma subtypes. In Figure 1B, IHC was used and graded SMARCB1 as 0 or 1 (absent or present). In pathology, few markers are just absent or present, and I am missing here at least an intermediate category with “half expression”. Or perhaps a quartile expression grading (Q1/Q2/Q3/Q4), or similar. Still, the number of cases analyzed (Figure 1C) is limited to draw strong conclusions about the prevalence of low SMARCB1 expression in lymphomas from younger vs older individuals. In Figure 1D, only a few cases are SMARCB1 null, which is true that are aged <20 years, whole only 1 case is aged 80. Again, the number of cases is not large enough to delineate a new entity of PTCL-NOS with SMARCB1 null expression in younger adults.

In figure 3, scRNAseq was performed in total spleen cells from mice with T-cell lymphoma. Histopathological studies are briefly shown as supplementary figures, but a pathological diagnosis of T-cell lymphoma needs to be confirmed. These mice carry homozygous deletion of Smarcb1 in Cd4 T cells driven by a specific cre-recombined in the Cd4 gene promoter. As controls, scRNA-seq data of mouse healthy spleen was retrieved from the Tabula Muris Consortium. I am not sure that these data can be used as a valid control, particularly to study T-cells in the TME. A much better control would have been cells from Cd4-cre mice. This is a very important concern for the correct interpretation of scRNAseq results.

Figure 3F and 3G. For interpretation and calculation of the % of the specific TME subpopulations in control vs tumor groups, the number of tumor cells should be excluded from the comparissom. If not, given that 44% of the total cells correspond to tumor cells in the lymphoma groups vs 0% in the healthy group, the % of the remaining subpopulations cannot be compared properly. In this regard, the composition of the TME in the lymphoma spleens and healthy mice could be defined by simple flow cytometry analyses. This would give as a second estimate of the changes in selected TME populations and validate data inferred from scRNAseq.

Results from scRNAseq data related to cell-cell interactions are shown in Figure 4B. I sincerely do not understand the arrows that connect the different cell populations. Complementary data shown as supplementary material should be integrated together to achieve a clearer vision describing the cellular/functional changes in the complex TNE. Again, some validation studies regarding potential expression/secretion of inflammatory cytokines (Il15, Il1b, or Ifng), immunosuppressive cytokines (Tgfb and Il1), or other chemokines (Ccl3, Ccl4 or Mif) could be performed, either by ELISA in serum or in tissue samples directly. Also, some ex vivo experimental studies could confirm the proposed direct:indirect tumor:TME cell interactions. I am positive that the information is there, and could be of sum interest, but more work needs to be carried out and a much better way of presenting results is required.

Regarding SAHA treatment, is it more effective in T-cell lymphoma cells with SMARCB1 null expression vs. those with expression? Was SAHA therapy effective to treat mouse lymphomas in vivo? Given that SAHA is currently in clinical use for treating patients with T-cell lymphomas, more experiments should be performed to demonstrate a selective therapeutic benefit. Otherwise, the author's claim regarding SAHA as a selectively effective drug for treating PTCL-SMARCB1- lymphomas could not be sustained.

Minor comments

Overall, the quality of the figures should be improved.

Page 6: The sentence, "Thus, T-PLL does not appear to be the human counterpart to the mature T-cell lymphomas in the Smarcb1-deficient mouse model" is not clear to me and requires an explanation. Figure 1, some legends are missing.

Reviewer #4 (Remarks to the Author): Expert in lymphoma clinical research and therapy, and vorinostat therapy

Dear Dr. Kerl,

I appreciate the chance to review this interesting research, and I applaud the thorough study and description of what may be a novel subset of PTCL-NOS.

Major Point:

My main question is how PTCL marked by SMARCB1 loss differs appreciably from PTCL in which other members of the SWI/SNF complex are altered. Collectively SWI/SNF alterations are fairly common in PTCL as assessed by sequencing but may be even more prevalent given the epigenetic silencing of SMARCB1 was found in to be the mechanism of loss in many (most?) of the cases in this series. I think this work would be significantly stronger if it were comprehensively profiling PTCL which features SWI/SNF alterations (ARID1A/1B/2, SMARCA2/4) including loss via epigenetic silencing to see if those collectively form a unique subset. The age distribution is provocative but not proof that this is a separate

subtype.

Minor Points:

Introduction

- Mutations in genes that form the SWI/SNF complex have been recurrently identified in NHL (even if SMARCB1 loss specifically is rare). This should be noted in the introduction.
- I would also add that SMARCB1 inactivation in mice also can lead to rhabdoid tumors in addition to T-cell lymphoma -- can cite reference from Roberts CW, Leroux MM, Fleming MD, Orkin SH. Highly penetrant, rapid tumorigenesis through conditional inversion of the tumor suppressor gene Snf5. Cancer Cell. 2002 Nov;2(5):415-25.

Methods

- Would mention PD1 levels too if this could be obtained even if not noteworthy (ideally along with the other immune checkpoints if possible) given their relevance in an exhausted TME

Discussion

- No mention of the T-PLL work is mentioned in the discussion. I would at least mention it once just to tie in to the nice correlative studies that were performed. Consider mentioning that that while 22q deletion is a common recurrent change in T-PLL, only 2 of the 16 cases had SMARCB1 loss and protein expression / methylation patterns did not suggest a major role for SMARCB1 iPLL pathogenesis.
- Line 425 -- the sentence "A key observation of our study was the 424 inverse correlation between myeloid and lymphoid infiltration in PTCL samples with." appears to be incomplete.
- I would mention that there have been multiple studies of vorinostat in PTCL and that other HDAC inhibitors are approved (romidepsin, belinostat).
- Given the immune exhaustion seen in the TME (CTLA4 specifically was mentioned) would checkpoint inhibition be a potential mechanism to test in this subset? Probably worth a sentence hypothesizing this point.

Point by point responses to referee comments:

Fischer et al., “Lack of SMARCB1 expression characterizes a subset of human and murine peripheral T-cell lymphomas”

We thank the reviewers for their appreciative evaluation of our study and their comments, which have helped to substantially improve the quality of the manuscript. By implementing all suggested changes, we hope to have carved out the central messages in more detail.

We now included more patients and performed cross-species studies and new single-cell experiments to further underline the relevance of our findings. The table below gives an overview of the major additions:

New data	Reviewers point	Figure
Characterization of 5 additional SMARCB1-negative PTCL-NOS cases	Reviewer 1 point 2 Reviewer 3 point 1 Reviewer 4 point 1	Figure 1D-J
Transcriptomic profiling of 5 SMARCB1-negative cases	Reviewer 1 point 1	Figure 1F
scRNA-Seq of 5 SMARCB1-negative cases	Reviewer 1 point 3 Reviewer 2 point 3	Figure 3/4
Comparison murine and human SMARCB1-deficient lymphomas	Reviewer 2 point 1, 3	Figure 2/5
Multiplex IHC analysis of murine tumor microenvironment	Reviewer 3 point 4	Figure 5C/D
New analysis of T15 RNA-Seq data	Reviewer 1 point 3	Figure 6K
SMARCB1 IHC staining of 15 mycosis fungoides cases	Reviewer 2 point 2	Suppl Figure 3A
Gene expression and promoter methylation of SWI/SNF members	Reviewer 4 point 1	Suppl Figure 5D

Reviewer #1: Expert in lymphoma genomics, scRNAseq, and in vivo models

Moreno et. al. identified a subtype of peripheral T cell lymphoma (PTCL) that appears to be driven by loss of SMARCB1. Using human samples and a mouse model, they demonstrated that this resulted from DNA hypermethylation in malignant T cells. With scRNA-seq, they identified additional immunosuppression in the TME. Using an HDAC inhibitor identified in a viability drug screen, they could remodel this TME in the mouse model. The molecular mechanisms of PTCL development remain incompletely characterized and this study can potentially fill an important gap. However, I have several concerns outlined in detail below about methodology in the analysis and the conclusions reached. There are also multiple instances in the manuscript where results are inadequately explained/characterized. I would encourage the authors to walk the reader through each result carefully. A few examples of this are included below.

We thank the reviewer for this feedback and address the specific points below. Regarding the careful explanation of the results, we completely restructured the second part of the manuscript for increased readability.

1. The authors claimed that 'SMARCB1-negative PTCL-NOS is a novel molecular subtype of PTCL enriched in young patients'. How does SMARCB1 loss correlate with currently described PTCL-NOS subtypes PTCL-GATA3 and PTCL-TBX21? What evidence supports that this is a novel subtype that is mutually exclusive with currently established subtypes?

Response: We thank the reviewer for this thoughtful comment. In order to compare the SMARCB1-deficient subtype to currently described ones we first generated RNA expression data of three SMARCB1-negative and three SMARCB1-positive pediatric PTCL cases using a probe-based method and, additionally, performed single-cell RNA-sequencing of five SMARCB1-negative PTCL cases and evaluated the expression of marker genes. We compared genes typically associated with the two mentioned subtypes. However, using the available immunohistochemistry data and the gene expression data, no clear assignment of the pediatric subtypes could be achieved. This new data is now presented in the new **Figure 1F/G**. This supports the hypothesis that SMARCB1-deficient PTCLs might be a new molecular subtype. We now include this description in the manuscript (**page 6, line 10-13**).

2. Additionally, the number of samples examined is quite low to conclude that this subtype is associated with CAYA.

Response: We completely agree that the samples size of the SMARCB1-negative PTCL-NOS samples is low. However, these cases are enriched in young patients and PTCL-NOS is very rare in children and young adults making it extremely difficult to get a large cohort of patients. We now increased the cohort of SMARCB1-deficient lymphomas by additional five samples to

ten cases in total, which is already double the number as presented before. Of these five cases, four were younger than 25 years, one patient was 28 age years old (**Figure 1 D/E**). Although these cases were specifically selected for being SMARCB1 negative on protein level, which might present a bias compared to our initial cohort, cases were not selected for age.

Additionally, there were recently three SMARCB1-negative cases reported by Havens et al. at the Meeting of the Society for Pediatric Pathologists (see: Havens et al., https://spp.memberclicks.net/assets/docs/SPP%202023%20Fall%20Meeting%20Abstract%20Book_Final.pdf). All of these patients were young (6, 11, 9 years) and characterized by “cytologic pleomorphism with diffuse CD45 expression and at least some T-cell marker positivity”, which is in line with what we observed for our cases (new **Figure 1G**). We include the information on these cases in the discussion (**page 14, line 7-9**). Moreover, there is one published case of a 14-year old male with biallelic loss of SMARCB1 and a hematopoietic neoplasm with differentiation of multiple lineages, which we now describe and cite in the introduction (**page 4, line 24/25**).

In total, we found further evidence that SMARCB1 deficient lymphomas are enriched in young patients. We searched for lymphoid malignancies with SMARCB1 negativity in IHC analysis. Of the nine additional cases (1 published, 3 as abstract and 5 new) seven were below 25 years old.

This new data is now included in the manuscript (**Introduction page 4, line 24, Results page 6, line 3-6, Discussion page 14, line 7-9, Figure 1D-J**).

In the discussion (line 416), the authors link PTCL with CHIP. Since CHIP is associated with increased age, how would this explain SMARCB1 loss as a driver in younger patients?

Response: We thank the reviewer for pointing this out. Indeed, the CHIP discussion is not directly linked to the age group where we identified SMARCB1 loss. However, CHIP is often identified in myeloid genes. Patients with SMARCB1 germline mutations/deletions develop rhabdoid tumors very early in life making it difficult to analyze the influence of germline SMARCB1 deficiency on lymphoid cells. However, in mosaicism patients there might be a role for SMARCB1 germline mutations in lymphoid cells. In this study, we did not find germline mutations but more sensitive methods to detect mosaicism at low percentages are developed at the moment (Fleischmann et al., submitted) and might clarify a potential role of SMARCB1 driving PTCL.

3. The scRNA-seq results are incompletely characterized.

Response: We regret that the scRNA-seq results appeared to be incompletely characterized in the earlier version of the manuscript. In the current version we have therefore tried to focus specifically on this aspect and, for example with regard to cell type annotation (see below), to choose a unified and simplified representation that still contains a sufficient level of detail. This applies to both the re-analysis of the earlier mouse data (new **Figures 5 and 7**; **Suppl. Figures 9, 10 and 12**), and the newly added scRNA-seq data from five human tumors (new **Figures 3 and 4**; **Suppl. Figures 7 and 8**).

- Please walk us through detailed cell type annotation, the choice of marker genes used (eg. how were MDSCs differentiated from macrophages?).

Response: We proceeded in the same way for cell type annotation in both cases. This was done in several steps. Below we provide an extensive overview of our approach. In addition, we have now described this procedure in detail in the **Suppl. Methods** section and refer to it at the appropriate points in the manuscript text.

The first step of cell type annotation always included the identification of cluster-specific DEGs (differentially expressed genes). Here we used not only log₂ fold-change (FC) as parameter, but also the “signal-to-noise” ratio (in other words: the selective expression of a specific gene), which is expressed by “delta_pct”, i.e. the difference of pct.1 (= percentage of cells expressing the corresponding DE gene in the examined cluster) minus pct.2 (= percentage of expressing cells in all remaining clusters; see **Tables S13, 19, 24**). This was followed by a manual inspection of the generated DEG lists to identify the most selective (i.e. large log₂ FC plus high delta_pct) genes as potential markers for a specific subpopulation of cells. Already in this first step, and based on existing knowledge from previous analyses (both our own and those of other groups), it is usually possible to quickly identify many TME populations defined by highly selective marker genes, e.g. immunoglobulin genes for B-cells, claudin 5 for endothelial cells. This was repeated when ambiguities occurred until either clear assignment was possible or these ambiguities were resolved, e.g. by subsetting and re-clustering of cognate meta-clusters (as was done for the Tumor/T-cell and Myeloid subsets in human PTCL; see **Figure 3D, E**) or by subclustering of individual clusters, which was then followed by another DEG analysis.

In addition, we also used publicly available databases (e.g. NCBI Pubmed, Gene), public domain tools (e.g. ToppGene Suite, STRING) and various search engines in order to classify genes with unknown functions and/or cell type specificity. Cluster 18 of human PTCL tumors can serve as a good example: it was initially identified as a myeloid/macrophage cluster using pan-specific markers such as ITGAM (CD11b) or CD68. The genes OSCAR and SIGLEC15, which were found to be highly selective in the DEG analysis, were subsequently identified as

specific markers for myeloid-derived osteoclasts through targeted Internet searches (**Suppl. Figure 7A, C**).

Finally, we used publicly available reference data sets including scRNA-seq cell atlases to achieve even more precise cell type mapping. Some examples: for in-depth annotation of stromal lymph-node NHC we used data from ref. 35 (**Suppl. Figure 7C**); for functional annotation of human PTCL-associated T-cells we used data from ref. 37 and references therein (**Figure 3L**); for the analysis of newly clustered spleens of WT and PTCL mice (**Figure 5; Suppl. Figure 9**) we used a spectrum of different reference data sets, among others Liang et al., 2023, PMID: 37944382; Kimmel et al., 2019, PMID: 31754020; or for a detailed annotation of the murine myeloid/neutrophil compartment Xie et al., 2020, PMID: 32719519.

Regarding the question of how we differentiated MDSC from e.g. macrophages: here too, we essentially proceeded as described above. Unlike many other specialized cell types, MDSC are defined at a functional level. Therefore, an accurate assignment of this cell type is naturally more difficult than in many other cases and it is therefore not surprising that there has been a long-standing debate about this very problem (see Bronte et al., 2016, PMID: 27381735). Nevertheless, we believe that through careful study of selected reference literature (single-cell studies included: Alshetaiwi et al., 2020, PMID: 32086381; Darden et al., 2021, PMID: 33021571; review articles included: Vanhaver et al., 2021, PMID: 34203451; Veglia et al., 2021, PMID: 33526920; Antuamwine et al., 2022, PMID: 35245287) as well as repeated checking of the extracted marker genes in our data, we were able to achieve a clear classification of human M-MDSC (**Figure 3M**) and murine PMN-MDSC (**Suppl. Figure 9A**) as well as a clear differentiation from other myeloid cell types.

In summary, our cell type annotation was performed through an interplay of bioinformatic analyses and careful manual curation. Due to lack of space or limitations regarding the total number of permitted citations, we cannot list the literature sources listed above for all analyses/cell type annotations carried out; nevertheless, we now describe the general procedure in much more detail than before in the **Suppl. Methods** section of the manuscript.

- ‘The cell distribution and sample composition are shown in Figure 6A-C and Suppl. Figure 11’ : This is an example where a description of the actual results would be highly beneficial to the reader.

Response: In our efforts to adhere to specified length restrictions for the manuscript, we have obviously exaggerated in some places, including in this case. In the current manuscript version, we have therefore tried to keep the text short and concise, but still sufficiently detailed for a good understanding of the results. In the specific case, which involved the description of the

quantitative and qualitative effects of TME remodeling by SAHA treatment in the mouse model (formerly Figure 6 and Suppl. Figure 11), the manuscript has undergone a comprehensive restructuring, which affects the presentation of the results in text and images, and we hope that this has significantly improved the readability of the manuscript and the comprehensibility of the results.

- 'Tumor cells most commonly interacted with Myeloid prog., monocytes and macrophages (Suppl Figure 9C)': how was 'most' determined?

Response: In the previous and current versions of the manuscript, this statement refers to the results of the InterCellar cell-cell interaction (CCI) analysis, in this specific case the evaluation of the significant CCIs between clusters of the TME and clusters of the tumor (previous Suppl. Figure 9C ; current version: **Suppl. Figure 10A**). The InterCellar tool, which we developed ourselves (Interlandi et al., 2021, PMID: 35017628), processes the interaction data from a previous CellPhoneDB analysis. In this specific case, the number of CCIs between clusters and/or compartments corresponds to that determined by CellPhoneDB based on the input data (i.e. the integrated scRNA-seq object from murine WT and PTCL spleen). However, for better clarity and congruence with the human interaction analysis (see new **Figure 4**), we have combined what were previously referred to as "Myeloid progenitors, monocytes and macrophages" together with other myeloid cell types such as neutrophils to form the higher-level compartment "Myeloid". It can be clearly seen from the two corresponding bar graphs in the new **Suppl. Figure 10A** that the Myeloid compartment with 2,633 total CCIs has a factor of 2.2 more interactions than the next highest T-/NK cell compartment (1,162 total CCIs) and even a factor of 11.2 more interactions than the B-cell compartment (234 total CCIs); similar size differences exist with regard to interactions with the tumor compartment, in which 158 Tumor-CCIs of the Myeloid compartment are a factor of 2.5 or factor 35 higher than those of the T-/NK cell compartment (62 Tumor-CCIs) or the B-cell compartment (8 Tumor-CCIs), respectively.

- What is the significance of the disappearance of tumor cluster 25 after SAHA treatment?

Response: This appears to be another example of a result which, to our regret, we apparently did not adequately explain in the earlier version. However, since we performed (partial) new clustering on the mouse samples and some of the corresponding figures either underwent major restructuring or were even omitted entirely (e.g. previous Suppl. Figure 12A), cluster 25-related results therefore disappeared from the manuscript. Nevertheless, we would like to do our best to answer the reviewer's question here. The complete disappearance of cluster 25

indicates a particularly high susceptibility of this tumor cell population to pan-HDACi by SAHA; yet, this observation remains at the descriptive level and any conclusion about it at the speculative level. At a quantitative level, we observed this effect (= reduction in tumor cell number under SAHA treatment) not exclusively for cluster 25 cells, but also, although less pronounced, for other populations within the tumor cell compartment. The different magnitudes of effects probably reflect to some extent the intratumoral heterogeneity of the tumor. Upon reexamining the upregulated DEGs of cluster 25, we now made the following observation: several important cell surface molecules including the three tumor necrosis factor receptors Tnfrsf4/9/18 as well as glycoprotein Cd160 are represented in the Top50 DEGs and are functionally linked to each other:

Left: Top50 DEGs of cluster 25

Right: STRING network of cluster 25 Top50 DEGs

Rank	Cluster_25_up	p_val	avg_log2FC	pct.1	pct.2	p_val_adj
1	Gcg	0	2,463	0,973	0,019	5,89E-308
2	S100a3	2,67E-273	2,152	0,936	0,067	5,18E-269
3	Epcam	5,02E-176	2,044	0,862	0,122	9,72E-172
4	S100a4	5,85E-133	1,794	1,000	0,351	1,13E-128
5	Fam162a	3,13E-213	1,763	1,000	0,315	6,07E-209
6	Mt3	2,83E-120	1,740	0,936	0,181	5,48E-116
7	Xcl1	3,519E-54	1,719	0,335	0,025	6,82E-50
8	Serpinb6b	1,17E-178	1,514	0,957	0,086	2,26E-174
9	Ernn	1,21E-185	1,507	0,995	0,127	2,34E-181
10	Arl6ip1	5,79E-170	1,468	0,995	0,692	1,12E-165
11	Mt2	1,98E-148	1,443	0,979	0,160	3,84E-144
12	Eno3	4,61E-180	1,410	0,989	0,124	8,93E-176
13	Casp3	2,01E-158	1,381	1,000	0,225	3,89E-154
14	Iitm2a	1,15E-143	1,322	0,984	0,178	2,23E-139
15	Sacs	6,89E-140	1,311	0,936	0,181	1,34E-135
16	Cenpu	2,74E-259	1,285	0,973	0,081	5,31E-255
17	Tnfrsf9	7,89E-133	1,266	0,995	0,205	1,53E-128
18	Lgals1	4,307E-92	1,264	1,000	0,503	8,348E-88
19	S100a10	2,253E-98	1,260	0,963	0,654	4,366E-94
20	Pkp4	1,32E-200	1,216	0,989	0,098	2,56E-196
21	Tnfrsf4	6,24E-133	1,208	0,989	0,185	1,21E-128
22	Mif	2,33E-103	1,167	1,000	0,638	4,52E-99
23	Mpl2	2,6E-178	1,163	0,979	0,097	5,04E-174
24	Etfb	2,1E-132	1,149	1,000	0,405	4,06E-128
25	Nkp7	3,46E-97	1,138	1,000	0,347	6,707E-93
26	Rps27l	8,26E-112	1,109	1,000	0,584	1,6E-107
27	Lat	1,38E-108	1,108	1,000	0,319	2,67E-104
28	Pmm1	1,11E-134	1,098	0,957	0,176	2,14E-130
29	Dcl1	3,242E-87	1,086	0,979	0,369	6,283E-83
30	Supt4a	2,27E-100	1,078	0,989	0,423	4,407E-96
31	Cd3g	1,291E-92	1,060	1,000	0,353	2,502E-88
32	Tesc	1,11E-115	1,057	0,846	0,119	2,15E-111
33	Nr4a2	6,93E-124	1,031	0,968	0,181	1,34E-119
34	Psmb9	2,6E-116	1,010	1,000	0,609	5,05E-112
35	Themis	6,72E-146	1,006	0,979	0,146	1,3E-141
36	Mllt3	3E-108	1,001	0,995	0,270	5,81E-104
37	Hsp90aa1	4,524E-95	1,000	1,000	0,782	8,768E-91
38	Cdkn2a	5,3E-141	0,995	0,979	0,161	1,03E-136
39	Ptpkr	9,06E-176	0,985	0,963	0,089	1,76E-171
40	Ikrf2	2,73E-128	0,984	0,979	0,180	5,3E-124
41	Lsm4	2,19E-110	0,971	1,000	0,599	4,25E-106
42	Gstt2	9,58E-119	0,970	0,973	0,204	1,86E-114
43	Pop5	3,556E-94	0,965	0,979	0,378	6,892E-90
44	Galnt7	1,79E-118	0,954	0,984	0,243	3,48E-114
45	Ezh2	3,429E-94	0,945	0,995	0,318	6,647E-90
46	Nol7	4,587E-95	0,943	0,989	0,598	8,89E-91
47	Tnfrsf18	1,11E-107	0,942	0,984	0,250	2,15E-103
48	Cd160	2,06E-148	0,937	0,846	0,071	4E-144
49	Arhgdig	1,55E-219	0,935	0,809	0,035	3E-215
50	Park7	2,221E-90	0,913	1,000	0,613	4,305E-86

These genes are part of a conserved intratumoral Treg signature that correlates with poor prognosis in multiple tumor types (Freeman et al., 2020; PMID: 32015231), and the authors suggested TNFRSF9 (aka 4-1BB) as a promising pan-cancer target. So to conclude here with another speculation: the complete disappearance of cluster 25 could represent the elimination of a particularly harmful subpopulation of tumor cells displaying a TNF receptor-related signature brought about by the pan-HDACi SAHA.

On a purely mathematical level, though, we can provide precise information about the significance of this observation. To this end, we performed a chi-square test to compare PCTL

and SAHA samples. This indicated high significance (p-value of 3.5e-89) for the disappearance of cluster 25 in SAHA-treated tumors (see graph below).

However, for the reasons mentioned above (= comprehensive restructuring of the manuscript), we have **omitted** all of these observations from the current version.

- What are the adjusted p-values for the genes identified in Fig. 7A?

Response: Unfortunately, the option of statistical testing is not available for the Seurat dot plot analysis used to create this figure (previous version: Figure 7A; new version: **Figure 7E**). We therefore attempted to determine the corresponding p-values using a customized DEG analysis. For this purpose, we selected SAHA samples as the first group and PTCL samples as second group and ran the “FindMarkers” function for the three cell types NK, NKT and T-cells and for the marker genes “Exhausted” and “Cytotoxic”, and display the result below in tabular form.

		SAHA_vs_PTCL: NK_cells				
	Gene_ID	p_val	avg_log2FC	pct.1	pct.2	p_val_adj
Exhausted	Btla	0,87975723	0,023	0,049	0,042	1
	Ctla4	1,6323E-07	-1,344	0,107	0,458	0,00316372
	Entpd1	0,21403504	0,219	0,061	0,000	1
	Havcr2	0,6426172	0,023	0,009	0,000	1
	Icos	0,44562012	-0,145	0,159	0,208	1
	Lag3	0,39429354	0,181	0,150	0,083	1
	Pdcd1	0,02037028	-0,250	0,015	0,083	1
	Tigit	0,36386071	0,100	0,034	0,000	1
	Tox	0,56734326	0,019	0,229	0,292	1
	Tox2	0,00328494	-0,297	0,021	0,125	1
	Vsir	0,92416381	-0,363	0,291	0,250	1
	Klrg1	0,00648928	1,085	0,245	0,000	1
Cytotoxic	Ccl5	6,2542E-09	3,848	0,813	0,208	0,00012122
	Gzma	2,5469E-07	4,585	0,639	0,042	0,0049364
	Gzmb	2,0797E-06	2,577	0,584	0,042	0,00403086
	Gzmk	0,34166991	0,319	0,098	0,042	1
	Prf1	6,9822E-05	1,611	0,431	0,000	1

		SAHA_vs_PTCL: NKT_cells				
	Gene_ID	p_val	avg_log2FC	pct.1	pct.2	p_val_adj
Exhausted	Btla	0,09951213	-0,078	0,055	0,110	1
	Ctla4	3,144E-20	-1,218	0,310	0,849	6,0938E-16
	Entpd1	0,04522577	-0,108	0,321	0,452	1
	Havcr2	6,1114E-05	-0,434	0,442	0,726	1
	Icos	0,00726674	-0,206	0,467	0,671	1
	Lag3	3,1078E-23	-1,204	0,215	0,781	6,0236E-19
	Pdcd1	7,839E-44	-1,703	0,095	0,849	1,5194E-39
	Tigit	0,06046133	-0,043	0,040	0,096	1
	Tox	2,7438E-37	-2,010	0,164	0,863	5,3181E-33
	Tox2	2,026E-16	-0,459	0,026	0,342	3,9268E-12
	Vsir	0,15590497	-0,027	0,336	0,452	1
	Klrg1	4,213E-09	0,955	0,536	0,151	8,1657E-05
Cytotoxic	Ccl5	0,01676105	-0,375	0,894	0,959	1
	Gzma	1,8032E-28	3,620	0,894	0,110	3,495E-24
	Gzmb	1,2967E-10	1,872	0,858	0,849	2,5133E-06
	Gzmk	0,00181367	-0,317	0,650	0,863	1
	Prf1	0,00028413	-0,411	0,496	0,685	1

		SAHA_vs_PTCL: T_cells				
	Gene_ID	p_val	avg_log2FC	pct.1	pct.2	p_val_adj
Exhausted	Btla	0,17830327	0,055	0,062	0,043	1
	Ctla4	2,8229E-19	-0,848	0,264	0,495	5,4713E-15
	Entpd1	2,7469E-07	-0,378	0,137	0,255	0,00532411
	Havcr2	0,02280806	-0,305	0,253	0,311	1
	Icos	0,01222204	-0,228	0,316	0,388	1
	Lag3	8,9561E-34	-1,230	0,107	0,363	1,7359E-29
	Pdcd1	4,9252E-58	-1,279	0,038	0,320	9,546E-54
	Tigit	1,1887E-06	-0,143	0,023	0,077	0,02303978
	Tox	5,3167E-80	-1,871	0,048	0,415	1,0305E-75
	Tox2	2,4591E-30	-0,387	0,009	0,135	4,7663E-26
	Vsir	0,07031205	-0,084	0,339	0,406	1
	Klrg1	8,9892E-18	0,903	0,356	0,111	1,7423E-13
	Cytotoxic	Ccl5	3,2899E-06	0,180	0,910	0,914
Gzma		6,255E-60	2,100	0,816	0,366	1,2123E-55
Gzmb		1,2487E-25	1,083	0,770	0,578	2,4203E-21
Gzmk		0,84538473	0,008	0,554	0,563	1
Prf1		0,00489714	-0,440	0,332	0,391	1

As can be seen, the corresponding p-values (unadjusted and/or adjusted) indicate non-significance for a number of these genes. The explanation for this lies in the different question and analysis method chosen: In dot plot analysis, the focus is on displaying different expression levels in different proportions of cell populations (in our case on selected, predefined marker genes), while DEG analysis is intended for unbiased identification of the most differentially expressed genes. However, despite the non-significance of many genes, the overall trend of SAHA in terms of the relative expression levels of exhaustion markers (down-regulation, blue color code) and cytotoxicity markers (up-regulation, red color code) is also evident here.

- In previous analysis, the authors demonstrated DNA hypomethylation in genes linked to myeloid cell differentiation with Smarcb1 deficiency. What was the effect of SAHA on myeloid gene expression?

Response: We took up this question and carried out an analysis of the existing RNA-seq data from the T15 system. This revealed that SAHA treatment of Smarcb1-deficient T15 PTCL cells leads to a statistically significant ($p = 1.6E-24$) upregulation of genes that are functionally involved in the differentiation of myeloid cells. We now show the associated functional gene network in **Figure 6K** and have also added this information to the manuscript text (**page 12, line 29**): “ In addition, the comparison between SAHA-treated and untreated T15 cells showed that HDACi leads to an upregulation of genes that are functionally involved in the differentiation of myeloid cells (**Figure 6K**). This implies a reversal of the epi-genotype of murine and human PTCL tumors, where we previously observed that genes particularly affected by DNA hypomethylation also include those of myeloid differentiation (cf. **Figure 2D**).“

- It appears that scRNA-seq with control and SAHA treatment was conducted in n=2 mice per group. If the authors have access to fixed/frozen samples from the mouse cohort, proportion and TME reprogramming findings could be validated eg. by IHC in some more samples.

Response: We thank the reviewer for this advice and have performed corresponding IHC validation experiments in which we confirm the main results of the comparative scRNA-seq analysis of the immune cell compartment in the spleen of WT and tumor-bearing mice, namely the significant decrease in B-cells and the significant increase in infiltrating, immunosuppressive myeloid cells (more specifically: Ly6g⁺ neutrophils) in the latter (new **Figure 5C**).

4. Drug screen: How were the 140 drugs chosen? How was SAHA chosen from these drugs? It would help the readers a lot if you could walk us through your results. ‘After validating different HDACi by cell viability, apoptosis and cell cycle assays, we selected SAHA for further experiments (Suppl. Figure 10B,C,D)’ – what do these results indicate compared to other drugs in the screen?

Response: Regarding question 1: Since SMARCB1, a core subunit of the SWI/SNF chromatin remodeling complex, is linked to epigenetic deregulation in various cancers, we were looking for a focused drug library with a balanced representation of a broad target spectrum of “epigenetically” active compounds. We ultimately chose the Cayman Chemicals “Epigenetic Modulators” library because it contains 140 different cell-permeable inhibitors against targets from the important classes of DNA methyltransferases (DNMT), histone methyltransferases and demethylases (HMT and HDM), histone acetyltransferases and deacetylases (HAT and HDAC) as well as acetylated histone-binding proteins (such as bromodomain proteins) (see **Table S23**).

Regarding question 2: In this drug screening, the Smarcb1-negative PTCL cell line T15 was particularly sensitive to the drug class of HDAC inhibitors (see **Suppl. Figure 11A**). We selected SAHA from this group because it is FDA-approved and we have shown in previous work that it can be used to control the growth of SMARCB1-negative rhabdoid tumor cells (Ref. 54: Kerl K et al., 2013, PMID: 23764045).

Regarding point/question 3: we have revised the corresponding passage regarding validation experiments of various HDACi in the current version. In addition, we have formulated the rationale for our selection of HDACi/SAHA more clearly and hope that we were able to answer the reviewer's questions and avoid any ambiguities for other readers (see **page 12, lines 4-15**).

Minor:

1. In the RNA-seq analysis of Smarcb1 re-expression (relating to Fig. 5B): details are all in the legend. Please mention in the text that you carried out RNA-seq. Are these results from only one experiment? The authors mentioned that ‘...viability and cell cycle were unaffected (not shown).’ Please include data on negative results. If neither of these were affected, is the reduction in cell growth a result of apoptosis? Could the authors explain and discuss this finding?

Response: As requested, we have added the key details of the bulk RNA-seq experiment to the manuscript text. This also applies to the question of the experiment/sample number; see **page 12**, from **line 22**: “We performed RNA-seq with three biological replicates each of untreated control T15 cells, SAHA-treated T15 cells, and Smarcb1-RE cells (Figure 6G).” Regarding the second request/question: We now show the associated „negative“ results/data, but can still only speculate why Smarcb1 reexpression in T15 cells leads to a decreasing cell number (**Figure 6F**) but has no significant influence on the apoptosis rate (**Suppl. Figure 11D**) or the cell cycle (**Suppl. Figure 11E**). Either the cell cycle has already settled back into a “normal” (i.e. unaltered) profile at this point. Alternatively, the absolute duration of one cycle passage could have increased without changing the relative proportions of the individual phases. However, we have omitted this speculation in the manuscript text and describe the results as follows (**page 12, lines 20-21**): “After re-expression, a significant reduction cell growth was observed (Figure 6F), while cell viability and cell cycle were not affected (Suppl. Figure 11 D,E).”

2. Please double check figure callout accuracy. Some of the instances with wrong callouts include - Line 317 “..by exhaustion markers such as Ctl4, Pdcd1 or Tox (Suppl. Figure 7A)’, should be Supplementary Figure 9A. Line 350 (Suppl. Figure 10A) should be Supplemental Figure 10B, Line 355: Figure 9F, should be Supplementary Fig. 10F

Response: This is, of course, a misfortunate oversight on our end and we now carefully checked all figure references.

3. Table S8: Tabula Muris links are broken. Please enter the actual numbers in the table. If these are unavailable, please describe the dataset in detail in the methods and how it was acquired and processed in the analysis.

Response: As suggested, we have now included all available numbers and information on the four scRNA-seq samples from our study (2x PTCL, 2x SAHA) as well as the two scRNA-seq

samples from the Tabula muris dataset (2x WT) in **Table S18**. We also describe in detail how these samples were processed bioinformatically in the Suppl. Methods in the two sections “Bioinformatic analysis of single-cell RNA-seq data” and “Reanalysis of publicly available datasets (murine control spleens)”.

4. There are instances of missing references. As an example: Line 330: ‘..T-cells are attracted to myeloid cells via Cxcl9/10, which then represses their function via..’

Response: We would like to thank the reviewer for pointing this out and have taken special care in the new version of the paper to ensure that these errors no longer occur.

5. Supplemental methods: ‘Due to the FFPE tissue, quality of the DNA was below average.’ What were the metrics and is the read quality reliable for CNV analysis? How was this determined?

Response: We apologize that the information was not clear enough and led to misunderstandings. The DNA quality was sufficient for SMARCB1 NGS and CNV analysis, however, we detected low quality scores in the DNA methylation analysis. This can often be observed in DNA from FFPE tissue. To avoid misunderstandings, we now clarify this aspect in the correct section of the methods part in the **Supplementary Methods**.

6. I might have missed this, but it does not seem like Supplemental Fig. 4B FISH results were described in the manuscript.

Response: We thank the Reviewer for pointing this out. Now we include the description of the FISH results in the manuscript on **page 6, line 17-18**; “In one case biallelic loss was confirmed using FISH (**Table S5; Suppl. Figure 4**).“ and in the **Methods** part.

7. Fig. 2D: this needs to be corrected for FDR.

Response: As suggested, we corrected the p-values for multiple testing using the Benjamini-Hochberg method. All FDR values were < 0.05 (range 0.0077 to 0.043) and are now listed in **Table S11**. For displaying reasons, we are still showing uncorrected p-values in **Figure 2D**.

Reviewer #2: Expert in T-cell lymphomas and genetically-engineered mouse models

The manuscript entitled 'Lack of SMARCB1 expression characterizes a subset of human and murine peripheral T-cell lymphoma (PTCL)' by Moreno et al. identified a subpopulation of PTCL-NOS young patients that lack SMARCB1 that currently have very poor outcome. They had previously developed a PTCL-NOS smarcb deficient mouse model and here they focus on the differences in immune cell phenotype from healthy tissues compared to malignant tissue (Tumor cells and TME) in these mice. They compared both in terms of methylation profiles and found a very strong similarity in methylation profiles with corresponding PTCL-NOS smarcb deficient PATIENTS in T cells and myeloid cells. Extensive single cell RNA expression analysis of the murine tumors showed a correlation of this methylation pattern with immune suppression, exhaustion and inflammation in the TME. After identification of methylation interfering drugs (histone deacetylase inhibitors) in vitro that were effective in inhibiting corresponding PTCL-NOS smarcb deficient cell lines, they used one of them 'SAHA' in their preclinical mouse model.

They showed again by single cell RNA expression data that the SAHA treatment completely reshaped the tumor microenvironment with CD8 T cells reverting from exhausted to effector cell phenotype, a stronger infiltration of NK and NKT and loss of tumor suppressive myeloid cells compared to untreated groups.

The reviewer agrees that these results argument for the use of pan-HDACi inhibitors in combination with other drugs for treatment of these specific subgroup of patients, which are in need of new therapeutic regimens.

I recommend this paper for publication but I have some comments though I would like to see addressed before

Comments to the authors

1) The preclinical mouse model used by the authors was previously developed (two publications) but it might be more convincing to give an extensive comparison between the PTCL-NOS SMARCB1- mouse tumor development and the counterpart patients in the introduction. Maybe a summarizing table comparing the immune phenotypes, pathology etc... in supplementary data might help a lot to convince the reader that these mice are valid as preclinical model for PTCL-NOS SMARCB1-patients.

Response: We thank the reviewer for this comment. However, the comparison of human SMARCB1-negative PTCL-NOS to murine Smarcb1-deficient lymphomas cannot be performed in the introduction as this is the first description of SMARCB1-deficient PTCL-NOS in humans. As suggested, we now extensively compare the characteristics of human und

murine SMARCB1-deficient lymphomas in **Figure 5**. Here, we show that the Smarcb1 deficient mice are a valid model to study PTCL-NOS.

2) Although the author show that PTCL-NOS patient are more frequently low in SMARCB1, quite some AITL patients also show lower expression; AITL though is a malignancy more touching older people; these AITL patient with low SMARCB1 are they younger than the overall AITL patient population and might they also develop a different methylation pattern and benefit from this treatment. Can the authors discuss this.

Response: We asked the same question during the preparation of this manuscript. Indeed, some patients of other T cell lymphoma subtypes also showed lower SMARCB1 expression (probably due to differences in tumor cell content), however, PTCL-NOS showed the most cases of low SMARCB1 expression. For both subtypes, PTCL-NOS and AITL, the expression of SMARCB1 did not show a correlation with age in this dataset.

To investigate the age correlation in other T cell lymphoma subtypes in more detail, we also collected a cohort of 15 mycosis fungoides (MF) patient with an age range of 22-87 years. Here, we did not detect any SMARCB1-negative cases. This data is now presented in **Suppl. Figure 3** and **Table S2**.

3) Related to the question 1, In figure 3G a lower level of B cells is detected in the PTCL spleens compared to controls spleens while myeloid populations are more prominent. Is this also true for human PTCL-NOS? the authors should show these data or mention is this is already previously published.

Response: Since we have now also examined human SMARCB1-negative PTCL-NOS tumors using scRNA-seq for the current manuscript, we can answer this question with a clear yes. Please see the new **Figure 5B**, which shows similar proportions of B-cell and myeloid compartments in mouse and human PTCL as well as similar trends (i.e. 7- to 8-fold decrease in B-cell numbers, 4- to 7-fold increase in myeloid infiltration) compared to WT spleens.

4) Figure 5. It might have been interesting to compare the differentially expressed genes not only in smarcb1 re-expressing cells versus untreated and SAHA treated but also cells without smarcb1 Knock-out to evaluate in how far SAHA treatment is reversing compared to a PTCL cell line with normal smarcb1 expression levels?

Response: Although we are grateful for the suggestion, we believe that the experiment is beyond the scope of our study. To our knowledge, there is only one PTCL cell line, T8ML-1 (Ehrentraut et al., 2017; PMID: 28659334). We did not have access to this cell line when planning and conducting the RNA-seq experiments. This cell line is derived from a 64-year-old

patient and has a highly altered karyotype and larger copy number alterations. Furthermore, it is not known whether T8ML-1 cells have a “normal” SMARCB1 expression level (however this level should be defined), and therefore, in our view, it is questionable whether this cell line would represent a good control or reference.

5) In Figure 6E, how do the authors explain the complete disappearance of the high number of B-cells in the tumor upon SAHA treatment.

Response: We probably have to explain this more thoroughly. The former Figure 6E, now **Suppl. Figure 12A**, does not show the complete disappearance of B-cells in the tumor after SAHA treatment, but the exact opposite, namely the repopulation of the tumor with diverse B-cell populations (FOB, MZB, GC, Progenitor B-cell like; see also previously Figure 6F, now **Suppl. Figure 12C**, for quantification).

6) As performed in Figure 4, can the authors also indicate the changes in crosstalk in terms of cytokines and chemokines which can explain the reduction of infiltrating myeloid cells and augmentation of T and NK more precisely in the TME?

Response: We regret that the design and content of the previous Figure 4B (“Schematic representation of TME and tumor cell communication...”) led to ambiguities, and not just for this reviewer, and have taken this as an opportunity to completely redesign it to include additional information, namely signature plots of selected signaling molecules that are involved in the functional crosstalk between the individual cell compartments (**Suppl. Figure 10C**). In our opinion, these diagrams clearly illustrate the changes in crosstalk that occur in murine PTCL tumors compared to healthy spleen tissue. However, we do not observe a „reduction of infiltrating myeloid cells“, but on the contrary a significant increase in myeloid infiltration (see new **Figure 5B**). Likewise, regarding T-/NK cells, we see an approximately 2-fold decrease (but no augmentation) at the quantitative level (from 20.6% in WT to 9.4% in PTCL). These quantitative changes are accompanied by strong changes in the functional profiles, namely a significant increase in T-/NK cell exhaustion and myeloid immunosuppressive features (new **Figure 5G**). We provide extensive context and a more precise explanation for the cell-cell communication axes in human PTCL (**page 9, from line 35**): “Of particular interest here are several chemokine signaling axes that are involved in TME remodeling through processes such as EMT and immunosuppression...”, and highlight this aspect again in the **Discussion (page 15, from line 13)**: „Main characteristics are diminished infiltration of T-cells and NK cells which at the same time have highly activated and exhausted phenotypes. This relationship has already been well described^{59,60,65}. Immunosuppressive cells of myeloid origin such as M-

MDSC and PMN-MDSC inhibit anti-tumor immune responses by impairing the activation and function of T- and NK cells^{61,66,67}.”.

7) Ethical statements on using patient data and performing mouse experiments should be indicated in the material and methods

Response: We agree that the Ethical statements should be easy to trace in the Materials & Methods section. Therefore, we now included a specific heading for this section (**page 17, line 16**).

Minor

1) Figure 3E Mistake on the axis ‘rogarnisation’ should be reorganization

Response: We thank the reviewer for pointing this out. Due to the integration of the human scRNA-seq data and the related restructuring of the figures, this figure is not anymore part of the manuscript.

2) Figure 4B Although this representation is clear the singly or double arrows are not clear and should be enlarged.

Response: We agree and have significantly modified the previous version of the figure and, among other additions such as associated signature plots, also enlarged the arrows in the new **Suppl. Figure 10C**.

Reviewer #3: Expert in lymphomas, genetically-engineered mouse models, and therapy

In this manuscript by Moreno and colleagues, a subgroup of PTCL with loss of SMARCB1 expression was identified, mostly affecting younger patients. Epigenomic and transcriptional sequencing in human PTCL-SMARCB1- lymphomas and in T-cell lymphomas developed in a mouse model with conditional deletion of Smarcb1 in Cd4 T cells, revealed similarities in mouse and human samples related to common pathways of lymphomagenesis. Analysis of the TME inferred from scRNAseq in murine spleen tumors revealed cellular and functional changes involving the lymphoma-microenvironment interaction network. Following a limited drug screen, HDAC inhibitors including SAHA showed increased anti-lymphoma activity against human T-cell lymphoma cells, and were able to remodel the cellular immune microenvironment in mouse T-cell lymphomas. The authors therefore claim that SHA could have a particular therapeutic value in PTCL with loss of SMARCB1.

Overall, the manuscript combines transcriptional and epigenomic sequencing studies in human samples and in a mouse T-cell lymphoma model driven by Smarcb1 loss. This is well written and provides new information, with potential clinical/therapeutic interest. I am however suggesting additional studies to be considered, which could strength the conclusions and improve the final quality of this work.

We thank the reviewer for this appreciative evaluation of our study.

Major concerns

1) The number of T-cell lymphomas samples analyzed in Figure 1 is limited.

Response: We agree that the number of cases is limited and addressed this concern as described in detail for point 2 of Reviewer 1. In brief, PTCL-NOS is very rare in children which might explain that so far only one case of biallelic SMARCB1 loss was reported in the literature (in a 14 years old male). We now increased the number of cases drastically (from n = 5 to n = 10) and additionally identified three cases which were presented recently confirming the importance of this subtype and reinforcing that SMARCB1 loss is associated with young age of the patients. The 5 new cases were of age (14-28 years) and the cases presented in an abstract online were of age (6-11 years).

As we now were actively were searching for these cases, we separated the cases for Figures on age correlation (new **Figure 1D/E**) in the screening cohort with the 5 original cases and the extended cohort including the 5 new cases. Although the new cases were specifically selected for SMARCB1 loss (negative in IHC staining) age was not considered in the selection of the cases.

This new data is now included in the manuscript (**Introduction page 4, line 24, Results page 6, line 3-6, Discussion page 14, line 7-9, Figure 1D-J**).

Regarding Figure 1A, expression of PTCL-NOS is relatively lower than in AITL, but similar to NK/TCL and HSTL. These data are not clear to me, in part because I do not find in the figure legends the definition of the acronyms for the T-cell lymphoma subtypes.

Response: We thank the reviewer for pointing out this shortcoming in the legend of Figure 1A, now **Figure 1B**. We now included the definitions of the acronyms. We agree that SMARCB1 expression is also low in HSTL and some cases of NK/TCL. However, only PTCL-NOS showed a higher number of patients with reduced expression levels and was therefore selected as subtype for further analyses.

In Figure 1B, IHC was used and graded SMARCB1 as 0 or 1 (absent or present). In pathology, few markers are just absent or present, and I am missing here at least an intermediate category with “half expression”. Or perhaps a quartile expression grading (Q1/Q2/Q3/Q4), or similar.

Response: We completely agree with the reviewer. As the IHC data presented here was generated in different institutions with different scoring systems, it was not possible to combine them to create one coherent figure. We observed that the correlation between gene and protein expression was poor. This can be explained by the fact that in PTCLs the tumor cell content is low. For example, for AITL, approximately 80% of cells are non-tumor cells (Pritchett et al., 2022, PMID: 34230608). Therefore, we decided to look at protein level, where expression can specifically be analyzed in tumor cells. However, correlation to gene expression was not good. Therefore, we decided to use IHC to identify cases without any protein expression (labelled with “0”) for further analysis.

Still, the number of cases analyzed (Figure 1C) is limited to draw strong conclusions about the prevalence of low SMARCB1 expression in lymphomas from younger vs older individuals. In Figure 1D, only a few cases are SMARCB1 null, which is true that are aged <20 years, while only 1 case is aged 80. Again, the number of cases is not large enough to delineate a new entity of PTCL-NOS with SMARCB1 null expression in younger adults.

Response: To address this point, we now added additional cases. In total, we now included nine additional cases, which were proven to be negative for SMARCB1 in IHC. This included one published case, three cases mentioned in an abstract and five cases that will additionally be published in this manuscript (described in more detail above). The age range of these cases was 6-28 years with seven patients being younger than 25 years.

2) In figure 3, scRNAseq was performed in total spleen cells from mice with T-cell lymphoma. Histopathological studies are briefly shown as supplementary figures, but a pathological diagnosis of T-cell lymphoma needs to be confirmed.

Response: The T cell lymphomas were extensively characterized in Roberts et al. (2002) and Wang et al. (2011). They showed that 100% of the mice developed mature CD8+ T cell lymphomas positive for Thy1.2, Cd3 and CD8 and negative for CD4, Tdt, B220, Mac1 and Gr1. Additionally, they found monoclonal rearrangement in all 20 mice analyzed (Roberts et al.). In more detail, Wang et al defined the cell of origin as a CD44^{hi}CD122^{lo} IL-15-independent subset of CD8 positive memory T cells when using the CD4-Cre line (n = 16). Thus, the Smarcb1-deficient mouse model (Snf5^{-/-}) was shown to effectively model human PTCL-NOS (Cutucache et al., 2016, PMID: 27725924). Therefore, we here only performed histopathological studies to confirm this diagnosis.

These mice carry homozygous deletion of Smarcb1 in Cd4 T cells driven by a specific cre-recombined in the Cd4 gene promoter. As controls, scRNA-seq data of mouse healthy spleen was retrieved from the Tabula Muris Consortium. I am not sure that these data can be used as a valid control, particularly to study T-cells in the TME. A much better control would have been cells from Cd4-cre mice. This is a very important concern for the correct interpretation of scRNAseq results.

Response: We thank the reviewer for this thoughtful comment and agree that Cd4-cre would also have been a valid control for scRNA-seq analysis of murine PTCL. However, with the newly added scRNA-seq analysis of human SMARCB1-negative PTCL tumors, we have provided another valid reference for the mouse dataset and our comparisons have shown that the non-malignant T-cells in the TME are very similar in terms of their relative amounts (see **Figure 5B**) as well as in relation to their phenotypic characteristics, which were recognized as terminally exhausted ("Tex") in both systems (see **Figure 3L** for human and **Figure 5G/Figure 7E** for murine PTCL).

3) Figure 3F and 3G. For interpretation and calculation of the % of the specific TME subpopulations in control vs tumor groups, the number of tumor cells should be excluded from the comparison. If not, given that 44% of the total cells correspond to tumor cells in the lymphoma groups vs 0% in the healthy group, the % of the remaining subpopulations cannot be compared properly. In this regard, the composition of the TME in the lymphoma spleens and healthy mice could be defined by simple flow cytometry analyses. This would give as a

second estimate of the changes in selected TME populations and validate data inferred from scRNAseq.

Response: As far as the quantification of the individual cell populations is concerned, this can of course be done in different ways. Despite the reviewer's valid comment, we retained the approach originally chosen, namely the inclusion of tumor cells in the analysis. This allowed us, for example, to directly compare the immune cell populations in human and mouse tumors. However, to solve the dilemma when comparing tumor and healthy tissue, we took another approach. By now forming the ratio between the B-cell and the Myeloid compartment, we can further highlight the important aspect of the inverse correlation between B-cell depletion and myeloid infiltration (see pie charts in new **Figure 5B**). Additionally, this type of representation also helps to illustrate the effect of SAHA in treated PTCL mice (see pie charts in new **Figure 7A**). Regarding the latter point: We were unable to implement the reviewer's suggestion of flow cytometric evaluation, but we are of the opinion that our new multiplex IHC analysis (new **Figure 5C**), with which we confirm the scRNA-seq results, makes up for this.

4) Results from scRNAseq data related to cell-cell interactions are shown in Figure 4B. I sincerely do not understand the arrows that connect the different cell populations.

Response: We regret that the previous version of the figure remained unclear to the reviewer. As we already explained above in a question from Reviewer 2 about the same figure, we have restructured the entire figure for the current manuscript (= new **Suppl. Figure 10**), enlarged the arrows in panel C and expanded this panel with additional illustrations, namely signature plots showing selected signaling molecules from the different functional categories and illustrating their differential expression in WT vs. PTCL tumor samples. Additionally, we have added an extended explanation of the arrows in panel C to the figure legend. With this thorough revision, we hope that no ambiguities remain in the redesigned version.

Complementary data shown as supplementary material should be integrated together to achieve a clearer vision describing the cellular/functional changes in the complex TME.

Response: We were happy to take up this request in the course of restructuring the text and images and supplementing the manuscript with new data. This time, we tried to better combine the previously distributed data and present it more uniformly and in a coherent way. As for the cellular changes within the different sample groups, we now show them: (i) as an overview of higher level compartments comparing WT and mouse or human PTCL (new **Figure 5B**), (ii) as an overview comparing WT vs. PTCL vs. SAHA in the mouse model (new **Figure 7A**), (iii) for the same comparison but with a higher level of detail for the T-/NK cell compartment in the

restructured **Figure 7D**, and finally (iv) for the B-cell compartment in the restructured **Suppl. Figure 12C**. As for the functional changes between the two sample groups WT and PTCL in the mouse, we have now presented these in the form of a split violin plot (new **Figure 5G**) and hope we have succeeded in illustrating the fundamental remodeling of the immune landscape in the TME of PTCL in a clear manner.

Again, some validation studies regarding potential expression/secretion of inflammatory cytokines (Il15, Il1b, or Ifng), immunosuppressive cytokines (Tgfb and Il1), or other chemokines (Ccl3, Ccl4 or Mif) could be performed, either by ELISA in serum or in tissue samples directly.

Response: We agree with the reviewer that these experiments would have been valuable in elucidating further details of the extensively remodeled immune landscape in SMARCB1-negative PTCL tumors. As described above, we used multiplex IHC on the MACSima platform (Miltenyi Biotec) to describe this in tissue sections of mouse splenic PTCL tumors (**Figure 5C/D**). Unfortunately, due to the unavailability of suitable antibodies (i.e. suited for MACSima imaging cyclic staining), we were unable to collect any data on the expression of the proposed cytokines/chemokines.

Also, some ex vivo experimental studies could confirm the proposed direct:indirect tumor:TME cell interactions. I am positive that the information is there, and could be of sum interest, but more work needs to be carried out and a much better way of presenting results is required.

Response: We have tried to improve the presentation of the L-R interaction analyses in the current version, and of course hope that the new **Figure 4** (human) and the restructured **Suppl. Figure 10** (mouse) do justice to these efforts. As for the “ex vivo experimental studies” proposed by the reviewer to confirm “direct:indirect tumor:TME cell interactions”, it is not clear to us which methodology and which (cellular) systems should be used. We therefore refrained from doing this, also because it would have gone beyond the scope of the present study.

5) Regarding SAHA treatment, is it more effective in T-cell lymphoma cells with SMARCB1 null expression vs. those with expression? Was SAHA therapy effective to treat mouse lymphomas in vivo? Given that SAHA is currently in clinical use for treating patients with T-cell lymphomas, more experiments should be performed to demonstrate a selective therapeutic benefit. Otherwise, the author’s claim regarding SAHA as a selectively effective drug for treating PTCL-SMARCB1- lymphomas could not be sustained.

Response: We cannot provide an answer to the first question because, as explained above (see our response to point 4 of reviewer 2), we did/do not have any SMARCB1-expressing PTCL cell line(s) available that we could use for this purpose. However, SAHA treatment had a greater cytotoxic effect in the Smarcb1-negative cell line T15 than in seven Non-Hodgkin-Lymphoma cell lines (see **Figure 6A**; **Suppl. Figure 11B**). Regarding the second question: SAHA treatment was not effective in treating lymphoma *in vivo* in terms of killing tumor cells and increasing the survival rate of the mice. However, as we demonstrate in **Figure 7** and **Suppl. Figure 12**, SAHA treatment of murine PTCL-NOS^{Smarcb1-} resulted in a reduction in myeloid infiltration with a concomitant increase in the cytotoxic cell profile and a decrease in the exhaustion phenotype. This observation suggests that SAHA would be suitable for the treatment of SMARCB1-negative PTCL patients in therapeutic approaches that involve combination with other agents such as e.g. immune checkpoint inhibitors. We take up exactly this point in the discussion (**page 16, line 10-12**), and this brings us to the reviewer's third point: in the new version of the manuscript we do not claim that "SAHA is a selectively effective drug for treating PTCL-SMARCB1". We believe that, due to the properties we have described, SAHA should neither be seen nor used as a "selectively effective" single agent, but rather in a combination therapy aimed at converting immunologically "cold" tumors into "hot" ones. This could be successful for SMARCB1-negative PTCL-NOS as well as other tumor entities with a similar profile. In the present study, with the molecular and cellular description of this aggressive PTCL subtype, we have laid the foundation for the reviewer's proposed additional and future experiments in this direction.

Minor comments

Overall, the quality of the figures should be improved.

Response: We have placed a specific focus on exactly this aspect in the current version of our study, and hope that the reviewer agrees with us that the new and revised illustrations are of higher quality.

Page 6: The sentence, "Thus, T-PLL does not appear to be the human counterpart to the mature T-cell lymphomas in the Smarcb1-deficient mouse model" is not clear to me and requires an explanation.

Response: We explain this now in more detail and also refer to T-PLL in the discussion.

Figure 1, some legends are missing.

Response: We added the missing information.

Reviewer #4: Expert in lymphoma clinical research and therapy, and vorinostat therapy

Dear Dr. Kerl,

I appreciate the chance to review this interesting research, and I applaud the thorough study and description of what may be a novel subset of PTCL-NOS.

Major Point:

My main question is how PTCL marked by SMARCB1 loss differs appreciably from PTCL in which other members of the SWI/SNF complex are altered. Collectively SWI/SNF alterations are fairly common in PTCL as assessed by sequencing but may be even more prevalent given the epigenetic silencing of SMARCB1 was found in to be the mechanism of loss in many (most?) of the cases in this series. I think this work would be significantly stronger if it were comprehensively profiling PTCL which features SWI/SNF alterations (ARID1A/1B/2, SMARCA2/4) including loss via epigenetic silencing to see if those collectively form a unique subset. The age distribution is provocative but not proof that this is a separate subtype.

Response: We now looked at gene expression data and promoter methylation of ARID1A, ARID1B, ARID2, SMARCA2 and SMARCA4 (**Suppl. Figure 5**). In contrast to SMARCB1, what was found lower expressed in PTCL-NOS than in AITL, the other SWI/SNF members showed no clear difference between these two subtypes. However, SMARCB1 was found at lower expression levels in HSTL but HSTL shows higher expression levels of ARID2, ARID1B and SMARCA2 compared to the other subtypes. For promoter methylation, only ARID2 and ARID1B showed increased methylation levels similar to SMARCB1. The other SWI/SNF members did not show higher promoter methylation in a specific subtype.

To strengthen the age distribution, we included additional SMARCB1-deficient cases as described above.

Minor Points:

Introduction

- Mutations in genes that form the SWI/SNF complex have been recurrently identified in NHL (even if SMARCB1 loss specifically is rare). This should be noted in the introduction.

Response: We thank the reviewer for this suggestion. We now included additional information on mutations in genes that are part of the SWI/SNF complex on **page 4, line 13-14**: "Chromatin remodeling genes such as SMARCA4, ARID1A, and other members of the SWI/SNF complex, can also be mutated in different lymphomas". Here, we now, amongst others, cite two studies which recently described the role of ARID1A and SMARCA4 in lymphomagenesis (Barisic et

al., 2024; Deng et al., 2024). Additionally, we now mention a case report describing SMARCB1 biallelic loss in a multi-lineage hematopoietic malignancy (Kinnaman et al., 2020).

- I would also add that SMARCB1 inactivation in mice also can lead to rhabdoid tumors in addition to T-cell lymphoma -- can cite reference from Roberts CW, Leroux MM, Fleming MD, Orkin SH. Highly penetrant, rapid tumorigenesis through conditional inversion of the tumor suppressor gene *Snf5*. *Cancer Cell*. 2002 Nov;2(5):415-25.

Response: We added this information in the introduction on **page 4, line 25-27**: “In a genetic mouse model (CD4-Cre *Smарcb1*^{fl/fl}) *Smарcb1* inactivation in mature T-cells triggers the development of oligoclonal TdT⁻, TCR⁺, CD3⁺, CD8⁺, and CD4⁻ mature PTCL (Wang et al., 2011) and, rarely, also rhabdoid tumors (Roberts et al., 2002).”

Methods

- Would mention PD1 levels too if this could be obtained even if not noteworthy (ideally along with the other immune checkpoints if possible) given their relevance in an exhausted TME

Response: We don't quite understand where in the Methods section we should place this information. However, in multiple figures we show the expression levels of the genes encoding PD-1 in connection with signatures for T-/NK cell exhaustion: (i) the human *PDCD1* gene in the T_{pex} signature within the Tumor/T-cell subset in the violin plot of **Figure 3L**, (ii) the murine *Pdcd1* gene in the “Exhausted” signature of PTCL samples in the split violin plot in **Figure 5G**, (iii) in T_{ex} clusters of the T-/NK cell compartment (**Figure 7C**), (iv) in the comparative dot plot analysis in **Figure 7E** and finally (v) in the STREAM trajectory in **Figure 7F**. We firmly believe that this provides sufficient information regarding relative PD-1 expression levels in exhausted versus normal T-/NK cell populations.

Discussion

- No mention of the T-PLL work is mentioned in the discussion. I would at least mention it once just to tie in to the nice correlative studies that were performed. Consider mentioning that that while 22q deletion is a common recurrent change in T-PLL, only 2 of the 16 cases had SMARCB1 loss and protein expression / methylation patterns did not suggest a major role for SMARCB1 iPLL pathogenesis.

Response: We now discuss the role of SMARCB1 loss in T-PLL in the discussion on **page 14, line 22-24**: “By analyzing SMARCB1 RNA and protein expression levels in multiple subtypes of mature T-cell lymphomas, we could exclude T-PLL, MEITL, EATL, MF, AITL and ALK-negative ALCL being the human counterpart of the phenotype observed in *Smарcb1*-deficient mice.”

- Line 425 -- the sentence "A key observation of our study was the 424 inverse correlation between myeloid and lymphoid infiltration in PTCL samples with." appears to be incomplete.

Response: We have revised the relevant passage in the Discussion (see **page 15, line 7-9**: "We were also able to observe similar patterns when comparing the relative proportions of higher-level cell compartments in human and mouse tumors. One of these patterns is the inverse correlation of myeloid and lymphoid (in particular B-cell) infiltration in PTCL.").

- I would mention that there have been multiple studies of vorinostat in PTCL and that other HDAC inhibitors are approved (romidepsin, belinostat).

Response: We now discuss the use of HDACi in T-cell malignancies in more detail on discussion:

Page 15, line 26-30: "Efficacy of several HDACi including SAHA is described for various hematological neoplasms (Eckschlager et al., 2017). SAHA is FDA-approved and in clinical use for relapsed or refractory (R/R) cutaneous T cell lymphoma (CTCL) with tolerable toxic effects (Hummel et al., 2013). Romidepsin, a selective HDAC1 and 2 inhibitor, and belinostat, a broad-spectrum HDACi, are FDA-approved for R/R PTCL (Lu et al., 2023)."

- Given the immune exhaustion seen in the TME (CTLA4 specifically was mentioned) would checkpoint inhibition be a potential mechanism to test in this subset? Probably worth a sentence hypothesizing this point.

Response: We agree that this is a very interesting aspect. We included a sentence speculating on this point in the discussion on **page 16, line 10-12**: "As there is evidence for the reversibility of CD8+ T-cell exhaustion after immune checkpoint blockade, checkpoint inhibitors might be a potential treatment option for these patients in future combinatorial clinical studies."

We hope we responded adequately to all points raised.

REVIEWERS' COMMENTS

Reviewer #1 (Remarks to the Author):

All my concerns have been adequately addressed. I appreciate the effort that has gone into this extensive revision.

Reviewer #2 (Remarks to the Author):

The authors have answered extensively and clearly to my comments and added multiple new data that support their findings

Reviewer #3 (Remarks to the Author):

The authors have responded to most of my criticisms and improved the quality of this nice manuscript. I can see that they have also responded to the other reviewers. They have performed more experiments, added more data and cases, and modified the figures.

Reviewer #4 (Remarks to the Author):

Thank you for addressing the questions thoughtfully. No further comments and I recommend publication.